# Orthogonal Transformer: An Efficient Vision Transformer Backbone with Token Orthogonalization

**Huaibo Huang**[1,2]**, Xiaoqiang Zhou**[1,2,4]**, Ran He**[1,2,3,*]
[1]Center for Research on Intelligent Perception and Computing, CASIA, Beijing, China
[2]National Laboratory of Pattern Recognition, CASIA, Beijing, China
[3]School of Artificial Intelligence, University of Chinese Academy of Sciences, Beijing, China
[4]University of Science and Technology of China, Hefei, China
huaibo.huang@cripac.ia.ac.cn, xq525@mail.ustc.edu.cn, rhe@nlpr.ia.ac.cn

## Abstract

We present a general vision transformer backbone, called as Orthogonal Transformer, in pursuit of both efficiency and effectiveness. A major challenge for vision transformer is that self-attention, as the key element in capturing long-range dependency, is very computationally expensive for dense prediction tasks (e.g., object detection). Coarse global self-attention and local self-attention are then designed to reduce the cost, but they suffer from either neglecting local correlations or hurting global modeling. We present an orthogonal self-attention mechanism to alleviate these issues. Specifically, self-attention is computed in the orthogonal space that is reversible to the spatial domain but has much lower resolution. The capabilities of learning global dependency and exploring local correlations are maintained because every orthogonal token in self-attention can attend to the entire visual tokens. Remarkably, orthogonality is realized by constructing an endogenously orthogonal matrix that is friendly to neural networks and can be optimized as arbitrary orthogonal matrices. We also introduce Positional MLP to incorporate position information for arbitrary input resolutions as well as enhance the capacity of MLP. Finally, we develop a hierarchical architecture for Orthogonal Transformer. Extensive experiments demonstrate its strong performance on a broad range of vision tasks, including image classification, object detection, instance segmentation and semantic segmentation.

## 1   Introduction

Recently, transformer [46] has made a tremendous success in the field of natural language processing (NLP). Benefiting from the self-attention (SA) mechanism, it has a strong capacity in building long-range dependencies in sequential data. Since long-range modeling is also essential for a wide range of vision tasks, transformer has been adapted into computer vision by converting an image into a sequence of patches [13] (called as tokens) and achieved competitive performance compared to CNN [43, 44]. Moreover, self-attention has a quadratic computational complexity to the number of tokens, resulting in intolerably expensive computational cost for dense prediction tasks, e.g., object detection and segmentation.

Many efforts have been made to design efficient self-attention mechanisms for vision transformer. PVT [50] employs a pyramid architecture with downsampled key and value tokens to reduce the computation of global attention. Swin [35] performs self-attention in a local region with shifted window to allow cross-region connection. GG-Transformer [63] and CrossFormer [52] present dilated

---

* Ran He is the corresponding author.

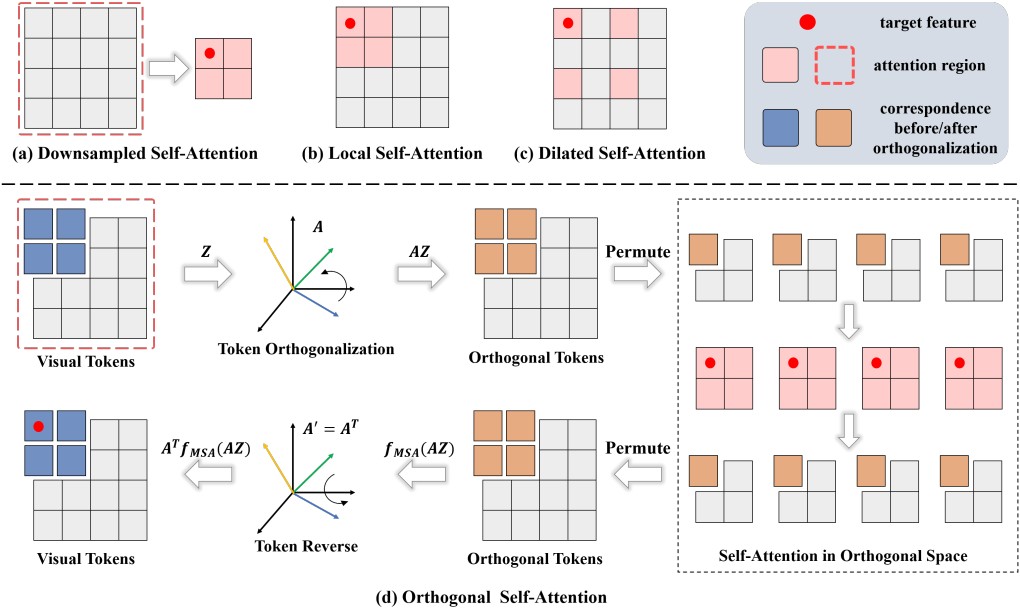

Figure 1: Illustration of different kinds of efficient self-attention mechanisms.

self-attention to learn large-scale features efficiently. As shown in Fig. 1 (a-c), the aforementioned attention mechanisms can reduce the number of tokens conveniently. However, the penalty is losing fine-level details for coarse global self-attention (including downsampled and dialted ones) or hurting long-range modeling for local self-attention [41, 52, 42, 63].

This paper presents an orthogonal self-attention (OSA) mechanism to capture global dependency without losing fine-level details. As shown in Fig. 1 (d), we orthogonalize tokens within local regions, permute them into token groups and perform group-wise self-attention in the orthogonal space. We apply Orthogonal Transformation (OT) for the tokens for the following reasons. Firstly, tokens have a lower resolution in the orthogonal space than in the original visual space, thus reducing the computational cost (details see Sec. 3.2.2). Secondly, the orthogonal space is reversible to the visual space without information loss (thus outperforming downsampled self-attention), and can be easily reversed back since the inverse matrix $\mathbf{A}'$ is just the transpose of the orthogonal matrix $\mathbf{A}$, i.e., $\mathbf{A}' = \mathbf{A}^{\mathrm{T}}$. Thirdly, OT can separate tokens into linearly independent groups. Thereby computing self-attention in such groups helps to explore different properties in representation. Lastly, OT builds connections among adjacent tokens explicitly and is more capable of modeling local correlations than dilated partition.

Despite of these benefits, it often needs complex optimization algorithms [31, 32] or imposing extra penalties on the loss functions [58, 48] to enforce orthogonality. In this work, we construct an endogenously orthogonal matrix that is friendly to neural networks with gradient optimizers. We build an orthogonal matrix $\mathbf{A} \in \mathbf{R}^{n \times n}$ by the product of $n$ Householder transformations $\{\mathbf{H}_i = \mathbf{I} - 2\mathbf{u}_i\mathbf{u}_i^{\mathrm{T}} | i = 0, .., n - 1\}$, where $\mathbf{u}_i \in \mathbf{R}^n$ is a learnable unit vector. In this way, $\mathbf{A}$ is naturally orthogonal and can be optimized as arbitrary orthogonal matrices. In particular, the dilated self-attention can be viewed as a degeneration of OSA when $\mathbf{A} = \mathbf{I}$. The strong capacity in exploring diverse transformations makes OSA more essential for the model to achieve competitive performance.

Based on the orthogonal self-attention mechanism, we propose a new Orthogonal Transformer in this work. As shown in Fig. 2, it follows the hierarchical design [50, 35] and serves as a general-purpose backbone for computer vision. We stack window self-attention (WSA) and orthogonal self-attention (OSA) alternatively to capture both global and local dependencies. Note that although our OSA is superior in preserving local details, combing it with WSA can further enhance Orthogonal Transformer. To improve the flexibility for arbitrary resolutions, we adopt convolutional position embedding to incorporate position information. Specifically, Positional MLP (PMLP) is introduced by equipping MLP with a depth-wise convolution (DConv) after the non-linear activation. Such a simple design not only enables the network to generate position information flexibly, but also enhance the capacity of MLP. Besides, it also allows for downsampling within the transformer block

with strided DConv. We empirically show that downsampling within the block can achieve better performance than outside.

Extensive experiments demonstrate the strong performance of Orthogonal Transformer on a wide range of vision tasks. For example, without extra training data or supervision (such as token labeling [26] and distillation [43]), our large model Ortho-L achieves 85.4 top-1 accuracy on ImageNet-1K image classification, surpassing the previous state-of-the-art Swin Transformer [35] by **+1.2** with similar model size and FLOPs. Our base model Ortho-B achieves 53.0 box AP and 45.9 mask AP on the COCO detection task, 49.8 mIOU on the ADE20K semantic segmentation task, surpassing the Swin Transformer counterpart by **+1.2**, **+1.2** and **+2.2**, respectively. Under a smaller setting of FLOPs, our Ortho-S even obtain larger performance gains, i.e., **+2.1** on ImageNet classification with $224 \times 224$ resolution, **+1.8** box AP, **+1.6** mask AP on COCO detection and **+4.0** mIoU on ADE20K segmentation.

## 2 Related Work

**Vision Transformers.** Transformer network is firstly proposed in the natural language processing (NLP) field and achieves superior performance in many NLP tasks [46, 27]. Recently, there emerges a trend towards incorporating the transformer into computer vision tasks, which are previously dominated by CNNs [19, 40]. ViT [13] is the pioneering work of vision transformers. After that, there are a large number of works focusing on designing a general vision Transformer backbone [9, 17, 35, 5, 50, 14, 66, 63, 61, 12, 52, 60, 26] for different vision tasks. To adapt transformer to the image inputs, hierarchical architectures [35, 50], efficient self-attentions [35, 63, 25] and diverse positional encodings [46, 38, 10, 12] are proposed. Vision transformers are also applied to different downstream vision tasks, such as object detection [4, 69], semantic segmentation [39, 67, 53], image restoration [6, 47], and video processing [33, 1]. We propose a new vision transformer backbone to tackle various vision tasks efficiently and effectively.

**Efficient Self-Attentions.** Many efficient self-attention mechanisms [30, 29, 8, 2, 49, 37] have been proposed in the field of NLP to efficiently handle long sequences. Since the image resolution is usually high in many vision tasks, the global self-attention can be applied only for a few times and in the low resolution feature space. Early approaches, such as ViT [13] and DeiT [43], employ large image patches to restrict the number of tokens. PVT [50] and Twins [9] use downsampled tokens to compute self-attention. Many works adopt local self-attention to limit the token number by attending only sub-regions of the input, such as the window attention [35], axis attention [21], and criss-cross attention [25]. Inspired by dilated convolution [62], GG Transformer [63] and CrossFormer [52] introduce dilated self-attention to capture long-distance dependency. Unlike existing methods, our orthogonal self-attention can attend to the entire tokens at a low computational cost. Global dependency can be captured without losing local details.

**Positional Encodings.** Recently, researchers propose to add different kinds of position information to the token feature or self-attention process. Absolute positional encoding (APE) [46] is the first work to add position encodings to the Transformer inputs. Relative positional encoding (RPE) [35, 38] considers the relative distance between two tokens instead of the absolute position. Conditional positional encoding (CPE) [10] takes feature as inputs and generates position information via a convolution layer. We follow CPE but compensate position information inside the MLP module. This design enables MLP to explore local correlations, thus enhancing the model performance.

**Orthogonality.** Orthogonality has been widely used in neural networks to regularize neurons [58, 24, 48] or learning disentangled representations [54]. In contrast to existing works, we employ orthogonality to group visual tokens for efficient self-attention. Besides, most of previous works depend on complex optimization algorithms, such as singular value decomposition [31] and iterative approximation [32], or imposing extra regularization terms on the loss functions [58, 48]. Unlike them, we construct an endogenously orthogonal matrix that is more friendly to neural networks with gradient optimizers.

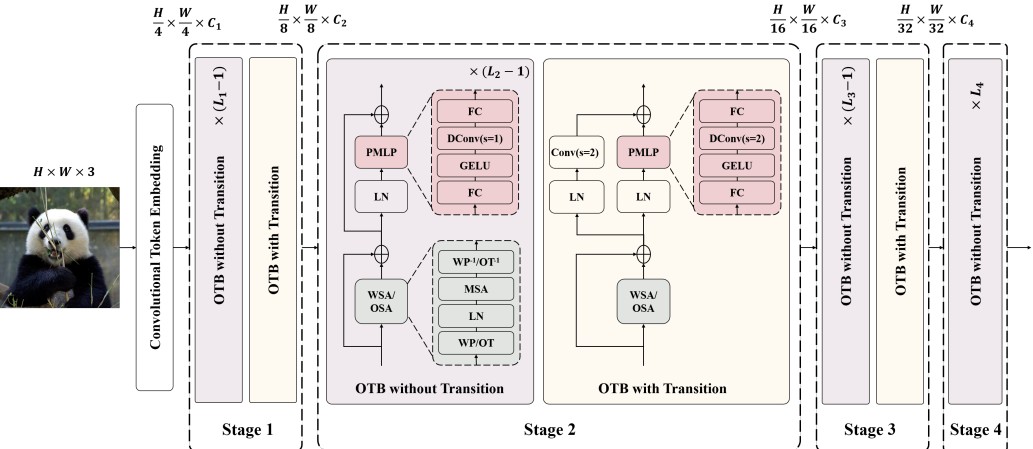

Figure 2: The architecture of Orthogonal Transformer. Orthogonal self-attention (OSA) and window self-attention (WSA) are used successively, where the tokens are grouped via window partition (WP) or orthogonal transformation (OT). WP$^{-1}$ and OT$^{-1}$ denote the respective reverse processes.

## 3  Method

### 3.1  Overall Architecture

An overview of the Orthogonal Transformer architecture is illustrated in Fig. 2. For an input image $x \in \mathbf{R}^{H \times W \times 3}$, we follow [57] and adopt several stacked convolution layers to obtain $\frac{H}{4} \times \frac{W}{4}$ patch tokens. Similar to Swin [35], the transformer blocks are divided into 4 stages to produce hierarchical representations. We stack blocks of window self-attention (WSA) and orthogonal self-attention (OSA) alternatively for better capacity of global and local modeling. For the Orthogonal Transformer blocks (OTBs), we introduce Positional MLP (PMLP) with an additional depth-wise convolution (DConv) after GELU in MLP to provide position information. Based on PMLP, we design two kinds of OTBs, one with transition and one without. OTBs with transition aim to reduce the spatial resolution by strided DConv and a residual strided convolution. The first three stages are composed of stacked OTBs without transition following by a single OTB with transition, while the last stage only consists of OTBs without transition.

### 3.2  Orthogonal Self-Attention

Orthogonal self-attention (OSA) is proposed to enable the transformer layers to encode high-resolution images efficiently. Unlike previous works [35, 50, 9, 63], OSA performs self-attention in the orthogonal space to capture global dependency without neglecting local correlations. It can cover the same global receptive field with much less cost than standard self-attention.

#### 3.2.1  Orthogonal Transformation

Orthogonal transformation (OT) corresponds to an orthogonal matrix $\mathbf{A}$ satisfying $\mathbf{A}^{\mathrm{T}}\mathbf{A} = \mathbf{I}$, where $\mathbf{I}$ is the identity matrix. As shown in Fig. 1, we employ orthogonal transformation $\mathbf{A}$ to convert visual tokens into orthogonal tokens for self-attention computation. Since it is hard to enforce orthogonality in neural networks [31, 32], we first introduce a simple yet effective way to construct an endogenously orthogonal matrix that can be optimized as arbitrary orthogonal matrices. It is based on the following linear algebra theorem [45] (Proof is provided in the appendix).

**Theorem 1.** *Every real orthogonal $n \times n$ matrix is the product of at most $n$ real orthogonal Householder transformations.*

Given $n$ tokens, inspired from Theorem 1, we construct an orthogonal matrix $\mathbf{A} \in \mathbf{R}^{n \times n}$ as:

$$\mathbf{A} = \mathbf{H_0}\mathbf{H_1}\cdots\mathbf{H_{n-1}}, \tag{1}$$

where $\mathbf{H}_i = \mathbf{I} - 2\mathbf{u}_i\mathbf{u}_i^{\mathrm{T}}$ is Householder transformation, $\mathbf{u}_i$ is a unit column vector. Providing a set of $n$ random-initialized vectors $\{\mathbf{v}_i | i = 0, ..., n-1\}$, the orthogonal matrix can be reformulated as:

$$\mathbf{A} = (\mathbf{I} - 2\frac{\mathbf{v}_0\mathbf{v}_0^{\mathrm{T}}}{\|\mathbf{v}_0\|^2})(\mathbf{I} - 2\frac{\mathbf{v}_1\mathbf{v}_1^{\mathrm{T}}}{\|\mathbf{v}_1\|^2})\cdots(\mathbf{I} - 2\frac{\mathbf{v}_{n-1}\mathbf{v}_{n-1}^{\mathrm{T}}}{\|\mathbf{v}_{n-1}\|^2}). \tag{2}$$

We set $\{\mathbf{v}_i | i = 0, ..., n-1\}$ as learnable parameters, and hence $\mathbf{A}$ can be optimized as arbitrary orthogonal matrices in neural networks with gradient optimizers. Remarkably, $\mathbf{A}$ can maintain orthogonality endogenously throughout the training process without extra regularizer.

### 3.2.2 OSA Block

To clarify, we first define two terms:

- **Orthogonal window size** $m_o$ is the size of the sub-window on which OT is performed. We perform OT separately for local windows to control its computation complexity.

- **Orthogonal groups** $n_o$ is the number of groups into which the tokens are separated by OT. Since the orthogonal matrix is square, $n_o = m_o^2$.

Now we elaborate how OSA works in the following three steps, token orthogonalization, self-attention for orthogonal tokens, token reverse.

**Token Orthogonalization.** Given the input feature $Z \in \mathbf{R}^{(h \times w) \times c}$ (viewed as $h \times w$ tokens with the dimension $c$), it is firstly divided into a grid of non-overlapped windows of size $m_o \times m_o$, i.e., $Z \in \mathbf{R}^{n_w \times n_o \times c}$, where $n_w = \frac{h}{m_o} \times \frac{w}{m_o}$ and $n_o = m_o^2$. $Z$ can be viewed as $n_w$ sub-windows, of which each consists of $n_o$ tokens, i.e., $Z = \{Z_i \in \mathbf{R}^{n_o \times c} | i = 0, \ldots, n_w - 1\}$. Then we perform OT separately for $Z_i$ by multiplying the orthogonal matrix $\mathbf{A} \in \mathbf{R}^{n_o \times n_o}$ and get orthogonalized tokens $\hat{Z}_i = \mathbf{A}Z_i \in \mathbf{R}^{n_o \times c}$. Finally, we combine the $n_w$ sub-windows and permute the feature $\hat{Z} \in \mathbf{R}^{n_w \times n_o \times c}$ into $n_o$ groups of orthogonal tokens $\hat{Z}^j \in \mathbf{R}^{n_w \times c}$ (where $j = 0, \ldots, n_o - 1$).

**Self-attention for orthogonal tokens.** We perform standard multi-head self-attention (MSA) group-wisely for $\hat{Z}^j$. Since $\hat{Z}^j$ has a smaller token number than the input $Z$ (actually $\frac{h}{m_o} \times \frac{w}{m_o}$ against $h \times w$), with an appropriate $m_o$, performing self-attention in the orthogonal space can significantly reduce the computation cost. Besides, for OSA, local correlations can be modeled because $\hat{Z}^j$ is computed from the adjacent tokens in $Z$, and local details are preserved due to the fact that $\hat{Z}$ and $Z$ are reversible to each other.

**Token reverse.** After the self-attention computation, we employ $\mathbf{A}^{\mathrm{T}}$, as the inverse of $\mathbf{A}$, to recover the visual tokens from the enhanced orthogonal representations. The token reverse process is not repeated herein because it is exactly the inverse process of token orthogonalization except that the orthogonal matrix is $\mathbf{A}^{\mathrm{T}}$.

The complete process of OSA can be defined as:

$$f_{OSA}(Z) = \mathbf{A}^{\mathrm{T}} f_{MSA}(f_{LN}(\mathbf{A}Z)) + Z. \tag{3}$$

We simplify the process of token orthogonalization as $\mathbf{A}Z$ for description convenience. Note that when $m_o = 1$, OSA equals to the global self-attention.

**Complexity Analysis.** The computation complexity of the standard global self-attention is

$$\Omega(GSA) = 4hwC^2 + 2(hw)^2C. \tag{4}$$

For our OSA, the computational complexity is

$$\Omega(OSA) = 4hwC^2 + \frac{1}{n_o}2(hw)^2C + 2n_ohwC. \tag{5}$$

The third term is produced by OT but can be ignored when $n_o \ll \sqrt{hw}$ (this is usually true for high-resolution vision tasks). Compared to the standard global self-attention, OSA's computation cost is reduced significantly from $O((hw)^2)$ to $O(\frac{1}{n_o}(hw)^2)$ for high-resolution vision tasks.

Table 1: Configurations of Orthogonal Transformer. The FLOPs are measured at resolution $224 \times 224$.

| Model | Blocks | Channels | Heads | Ratio | Params (M) | FLOPs (G) |
|---|---|---|---|---|---|---|
| Ortho-T | [2, 2, 6, 2] | [32, 64, 160, 256] | [1, 2, 5, 8] | 3 | 3.9 | 0.7 |
| Ortho-S | [3, 5, 13, 3] | [64, 128, 256, 512] | [2, 4, 8, 16] | 4 | 24 | 4.5 |
| Ortho-B | [3, 5, 19, 4] | [80, 160, 320, 640] | [2, 4, 8, 16] | 4 | 50 | 8.6 |
| Ortho-L | [4, 6, 24, 5] | [96, 192, 384, 768] | [3, 6, 12, 24] | 4 | 88 | 15.4 |

## 3.3 Window Self-Attention

Since Orthogonal Transformer is composed of stacked window self-attentions (WSAs) and OSAs, we give a brief review of WSA here. WSA first employ window partition (WP) to split the input feature $Z \in \mathbf{R}^{(h \times w) \times c}$ into $\frac{h}{m_w} \times \frac{w}{m_w}$ non-overlapped windows of size $m_w \times m_w$, then performs MSA within each window, and finally reconstruct the tokens from the enhanced representation. The complete process of WSA is defined as:

$$f_{WSA}(Z) = f_{WP}^{-1}(f_{MSA}(f_{LN}(f_{WP}(Z)))) + Z, \tag{6}$$

where $f_{WP}$ and $f_{WP}^{-1}$ denote window partition and its reverse, respectively. WSA has a linear computational complexity to token number $hw$:

$$\Omega(WSA) = 4hwC^2 + 2m_w^2 hwC. \tag{7}$$

## 3.4 Positional MLP

Earlier ViTs adopt absolute position embedding (APE) [46] or relative position embedding (RPE) [35, 38] to handle the position information but they cannot be well adapted for arbitrary input resolutions [10]. Recently, convolution position embedding (CPE) [10] is proposed to generate position information by convolutional layers. In detail, CPE adds position encodings into the input tokens outside the transformer blocks. We follow CPE [10] but apply it in a different way. As shown in Fig. 2, we introduce Positional MLP (PMLP) to tackle position information in the MLP module. We equip PMLP with a depth-wise convolution (DConv) after the non-linear GELU. This design can not only provide position information, but also enhance the capacity of MLP in exploring local correlations. Based on it, we further develop PMLP with transition by setting the stride of DConv as 2 and adding a residual strided convolution. PMLP without transition is defined as:

$$f_{PMLP}(Z) = f_{FC}(f_{DConv}^{(s1)}(f_{GELU}(f_{FC}(f_{LN}(Z))))) + Z, \tag{8}$$

while PMLP with transition is defined as

$$f_{PMLP}^{(t)}(Z) = f_{FC}(f_{DConv}^{(s2)}(f_{GELU}(f_{FC}(f_{LN}(Z))))) + f_{Conv}^{(s2)}(f_{LN}(Z)), \tag{9}$$

where $f_{DConv}^{(s1)}$ and $f_{DConv}^{(s2)}$ denote depth-wise convolutions with stride 1 and stride 2, respectively, and $f_{Conv}^{(s2)}$ denote a convolution with stride 2.

*Discussions.* Note that depth-wise convolution (DConv) has been applied in previous vision transformers. The major difference between the proposed Positional MLP and others is that we use strided DConv to perform downsampling within the transformer block. In this way, we do not need extra patch merging layer to perform feature downsampling. In Table 6, we empirically show that it can achieve better performance than outside the transformer block. We also conduct extensive experiments to investigate the best location of DConv in Appendix D.4. There exist various manners to employ DConv in vision transformers (e.g., PVT v2 [51], CeiT [64], CMT [16] and ours adopt DConv in different locations). We believe our work is suggestive to find appropriate manner of incorporating depth-wise convolution into transformers.

## 3.5 Architecture Variants

As shown in Table 1, we build four different Orthogonal Transformer backbones by changing the block number and channel number in each stage. Specifically, Ortho-S, Ortho-B and Ortho-L are designed to have a similar setting of FLOPs and model size with their Swin Transformer counterparts.

Table 2: Comparison with the state-of-the-art on ImageNet-1K classification.

| Model | #Param (M) | FLOPs (G) | Acc. (%) | Model | #Param (M) | FLOPs (G) | Acc. (%) |
|---|---|---|---|---|---|---|---|
| DeiT-Ti [43] | 5.7 | 1.3 | 72.2 | PVT-L [50] | 61 | 9.8 | 81.7 |
| PVTv2-b0 [51] | 3.7 | 0.6 | 70.5 | T2T-24 [65] | 64 | 13.2 | 82.2 |
| T2T-7 [65] | 4.3 | 1.1 | 71.7 | Swin-S [35] | 50 | 8.7 | 83.0 |
| QuadTree-B-b0 [41] | 3.5 | 0.7 | 72.0 | CvT-21 [55] | 32 | 7.1 | 82.5 |
| Ortho-T | 3.9 | 0.7 | **74.0** | CaiT-s24 [44] | 47 | 9.4 | 82.7 |
|  |  |  |  | Focal-S [61] | 51 | 9.1 | 83.5 |
| RegNetY-4G [36] | 21 | 4.0 | 80.0 | CrossFormer-B [52] | 52 | 9.2 | 83.4 |
| DeiT-S [43] | 22 | 4.6 | 79.9 | RegionViT-M [5] | 41 | 7.4 | 83.1 |
| PVT-S [50] | 25 | 3.8 | 79.8 | Ortho-B | 50 | 8.6 | **84.0** |
| T2T-14 [65] | 22 | 5.2 | 80.7 | CvT-21↑ 384 [55] | 32 | 25.0 | 84.9 |
| Swin-T [35] | 29 | 4.5 | 81.3 | CaiT-s24↑ 384 [44] | 47 | 32.2 | 84.3 |
| Twins-SVT-S [9] | 24 | 2.9 | 81.7 | Ortho-B↑ 384 | 50 | 26.6 | **85.2** |
| CvT-13 [55] | 20 | 4.5 | 81.6 | DeiT-B [43] | 86 | 17.5 | 81.8 |
| CaiT-xs24 [44] | 27 | 5.4 | 81.8 | Swin-B [35] | 88 | 15.4 | 83.3 |
| Focal-T [61] | 29 | 4.9 | 82.2 | CaiT-s48 [44] | 90 | 18.6 | 83.5 |
| CrossFormer-S [52] | 31 | 4.9 | 82.5 | Focal-B [61] | 90 | 16.0 | 83.8 |
| RegionViT-S [5] | 31 | 5.3 | 82.6 | CrossFormer-L [52] | 92 | 16.1 | 84.0 |
| Container [15] | 22 | 8.1 | 82.7 | RegionViT-B [5] | 73 | 13.0 | 83.2 |
| QuadTree-B-b2 [41] | 24 | 4.5 | 82.7 | Ortho-L | 88 | 15.4 | **84.2** |
| Ortho-S | 24 | 4.5 | **83.4** | Swin-B↑ 384 [35] | 88 | 47.0 | 84.2 |
| CvT-13↑ 384 [55] | 20 | 16.3 | 83.0 | CaiT-s48↑ 384 [44] | 90 | 63.8 | 85.1 |
| CaiT-xs24↑ 384 [44] | 27 | 19.3 | 83.8 | Ortho-L↑ 384 | 88 | 47.4 | **85.4** |
| Ortho-S↑ 384 | 24 | 14.3 | **84.8** |  |  |  |  |

For all the models, we set the orthogonal window sizes $m_o$ for OSA in the four stages as 8, 4, 2, 1, respectively, and the window size for WSA as 7. The expansion ratios in MLP are set as 3 for Ortho-T and 4 for the other three variants. For the convolutional patch embedding, we borrow early convolutions from [57] and apply 5 convolutions with the same setting of [57]. For the transformer blocks, we set the last one in each of the first 3 stages as OTB with transition. The second FC and residual convolution change the feature channels. The convolutions reduce the spatial resolution with the stride of 2. The kernel sizes of DConv and the residual convolution are set to be $5 \times 5$ and $3 \times 3$, respectively. More details are in the Appendix.

## 4 Experiments

We conduct experiments on a wide range of vision tasks: image classification on ImageNet-1K [11], object detection and instance segmentation on COCO 2017 [34], and semantic segmentation on ADE20K [68]. We also take ablation studies to validate the importance of each component. More details about experimental settings and extra ablation studies are in the Appendix.

### 4.1 Image Classification on ImageNet-1K

**Experimental Settings.** We benchmark our Orthogonal Transformer on ImageNet-1K [11] image classification. We follow the same training strategy in DeiT [43] and adopt the strong data augmentation and regularization, except for repeated augmentation [22] that does not improve performance. For a fair comparison, we do not use extra training data or extra supervision techniques, such as token labeling [26] and distillation [43], because most of previous works didn't use them. All our models are trained from scratch for 300 epochs with the input size of $224 \times 224$. An AdamW optimizer with a cosine decay learning rate scheduler and 5 epochs of linear warm-up is employed. The initial learning rate, weight decay, and batch-size are 0.001, 0.05, and 1024, respectively. For Ortho-T, we use a smaller weight decay of 0.01. The maximum rates of increasing stochastic depth [23] are set as 0.1/0.2/0.4/0.5 for the models from tiny to large. For the results of $384 \times 384$, we fine-tune the models for 30 epochs with learning rate of 1e-5, weight decay of 1e-8 and bach-size of 512.

Table 3: Object detection and instance segmentation with Mask R-CNN on COCO val2017. FLOPs are measured at resolution $800\times1280$. All the models are pre-trained on ImageNet-1K.

| Backbone | #Param (M) | FLOPs (G) | Mask R-CNN 1× schedule | | | | | | Mask R-CNN 3× + MS schedule | | | | | |
|---|---|---|---|---|---|---|---|---|---|---|---|---|---|---|
| | | | $AP^b$ | $AP^b_{50}$ | $AP^b_{75}$ | $AP^m$ | $AP^m_{50}$ | $AP^m_{75}$ | $AP^b$ | $AP^b_{50}$ | $AP^b_{75}$ | $AP^m$ | $AP^m_{50}$ | $AP^m_{75}$ |
| Res50 [20] | 44 | 260 | 38.0 | 58.6 | 41.4 | 34.4 | 55.1 | 36.7 | 41.0 | 61.7 | 44.9 | 37.1 | 58.4 | 40.1 |
| PVT-S [50] | 44 | 245 | 40.4 | 62.9 | 43.8 | 37.8 | 60.1 | 40.3 | 43.0 | 65.3 | 46.9 | 39.9 | 62.5 | 42.8 |
| Twins-S [9] | 44 | 228 | 43.4 | 66.0 | 47.3 | 40.3 | 63.2 | 43.4 | 46.8 | 69.2 | 51.2 | 42.6 | 66.3 | 45.8 |
| Swin-T [35] | 48 | 264 | 42.2 | 64.6 | 46.2 | 39.1 | 61.6 | 42.0 | 46.0 | 68.2 | 50.2 | 41.6 | 65.1 | 44.8 |
| ViL-S [66] | 45 | 218 | 44.9 | 67.1 | 49.3 | 41.0 | 64.2 | 44.1 | 47.1 | 68.7 | 51.5 | 42.7 | 65.9 | 46.2 |
| Focal-T [61] | 49 | 291 | 44.8 | 67.7 | 49.2 | 41.0 | 64.7 | 44.2 | 47.2 | 69.4 | 51.9 | 42.7 | 66.5 | 45.9 |
| RegionViT-S [5] | 51 | 183 | 44.2 | - | - | 40.8 | - | - | 47.6 | - | - | 43.4 | - | - |
| Ortho-S | 44 | 277 | **47.0** | **69.4** | **51.3** | **42.5** | **66.1** | **45.7** | **48.7** | **70.5** | **53.3** | **43.6** | **67.3** | **47.3** |
| Res101 [20] | 63 | 336 | 40.4 | 61.1 | 44.2 | 36.4 | 57.7 | 38.8 | 42.8 | 63.2 | 47.1 | 38.5 | 60.1 | 41.3 |
| X101-32 [59] | 63 | 340 | 41.9 | 62.5 | 45.9 | 37.5 | 59.4 | 40.2 | 44.0 | 64.4 | 48.0 | 39.2 | 61.4 | 41.9 |
| PVT-M [50] | 64 | 302 | 42.0 | 64.4 | 45.6 | 39.0 | 61.6 | 42.1 | 44.2 | 66.0 | 48.2 | 40.5 | 63.1 | 43.5 |
| Twins-B [9] | 76 | 340 | 45.2 | 67.6 | 49.3 | 41.5 | 64.5 | 44.8 | 48.0 | 69.5 | 52.7 | 43.0 | 66.8 | 46.6 |
| Swin-S [35] | 69 | 354 | 44.8 | 66.6 | 48.9 | 40.9 | 63.4 | 44.2 | 48.5 | 70.2 | 53.5 | 43.3 | 67.3 | 46.6 |
| Focal-S [61] | 71 | 401 | 47.4 | 69.8 | 51.9 | 42.8 | 66.6 | 46.1 | 48.8 | 70.5 | 53.6 | 43.8 | 67.7 | 47.2 |
| RegionViT-B [5] | 93 | 307 | 45.4 | - | - | 41.6 | - | - | 48.3 | - | - | 43.5 | - | - |
| Ortho-B | 69 | 372 | **48.3** | **70.5** | **53.0** | **43.3** | **67.3** | **46.5** | **49.9** | **71.4** | **54.8** | **44.3** | **68.6** | **47.9** |

Table 4: Object detection and instance segmentation with Cascade Mask R-CNN on COCO val2017.

| Method | #Params (M) | FLOPs (G) | 3× + MS schedule | | | | | |
|---|---|---|---|---|---|---|---|---|
| | | | $AP^b$ | $AP^b_{50}$ | $AP^b_{75}$ | $AP^m$ | $AP^m_{50}$ | $AP^m_{75}$ |
| DeiT [43] | 80 | 889 | 48.0 | 67.2 | 51.7 | 41.4 | 64.2 | 44.3 |
| Swin-T [35] | 86 | 745 | 50.5 | 69.3 | 54.9 | 43.7 | 66.6 | 47.1 |
| Focal-T [61] | 87 | 770 | 51.5 | 70.6 | 55.9 | - | - | - |
| Ortho-S | 81 | 755 | **52.3** | **71.3** | **56.8** | **45.3** | **68.6** | **49.2** |
| X101-32 [59] | 101 | 819 | 48.1 | 66.5 | 52.4 | 41.6 | 63.9 | 45.2 |
| Swin-S [35] | 107 | 838 | 51.8 | 70.4 | 56.3 | 44.7 | 67.9 | 48.5 |
| Ortho-B | 107 | 851 | **53.0** | **72.0** | **57.4** | **45.9** | **69.4** | **49.9** |

**Results.** Table 2 reports the image classification results on ImageNet-1K. It clearly shows that our Orthogonal Transformer has a stronger performance than previous models under similar settings of FLOPs and model size. Specifically, the tiny model Ortho-T achieves a 74.0% Top-1 accuracy with only 0.7G FLOPs, surpassing QuadTree-B-b0, T2T-7 and PVTv2-b0 by 2%, 2.3% and 3.5%, respectively. The small model Ortho-S achieves the same accuracy as base model CrossFormer-B with 51% fewer FLOPs. As for $384 \times 384$ input size, the large model Ortho-L achieves an accuracy of 85.4%, surpassing Swin-B by 1.2% with similar model size, and outperforming CaiT-s48 with 26% fewer FLOPs. These quantitative comparisons with SOTA methods demonstrate the efficiency and effectiveness of Orthogonal Transformer.

## 4.2 Object Detection on COCO

**Experimental Settings.** Experiments on object detection and instance segmentation are conducted on COCO 2017 dataset [34]. Following [35], we use our transformer models as the backbone network, and use Mask-RCNN [18] and Cascaded Mask R-CNN [3] as the detection and segmentation heads. For both tasks, the backbones are pretrained on ImageNet-1K, and fine-tuned on the COCO training set with AdamW optimizer. We take experiments on two common settings: "1 ×" (12 training epochs) and "3 × +MS" (36 training epochs with multi-scale training). The configurations follow the setting of [35] and are implemented with MMDetection [7].

**Results.** Table 3 and Table 4 show the results with Mask R-CNN and Cascade Mask R-CNN, respectively. The results show that our method achieves the best performance in all comparisons. For vision transformers with Mask R-CNN framework, our Ortho-S outperforms Focal-T (having similar model size and computational cost) by **+2.2** box AP, **+1.5** mask AP with the 1× schedule and **+1.5** box AP, **+0.9** mask AP with the 3× multi-scale learning schedule. The results with Cascade Mask R-CNN also show that our Orthogonal Transformer exceeds the counterparts by evident margins.

Table 5: Semantic segmentation with Semantic FPN and Upernet on ADE20K. The FLOPs are measured at resolution 512×2048.

| Backbone | Semantic FPN 80k | | | Backbone | Upernet 160k | | | |
|---|---|---|---|---|---|---|---|---|
| | #Param (M) | FLOPs (G) | mIoU (%) | | #Param (M) | FLOPs (G) | mIoU (%) | MS mIoU (%) |
| Res50 [20] | 29 | 183 | 36.7 | TwinsP-S [9] | 55 | 919 | 46.2 | 47.5 |
| PVT-S [50] | 28 | 161 | 39.8 | Twins-S [9] | 54 | 901 | 46.2 | 47.1 |
| TwinsP-S [9] | 28 | 162 | 44.3 | Swin-T [35] | 60 | 945 | 44.5 | 45.8 |
| Swin-T [35] | 32 | 182 | 41.5 | Focal-T [61] | 62 | 998 | 45.8 | 47.0 |
| Ortho-S | 28 | 195 | **48.2** | Ortho-S | 54 | 956 | **48.5** | **49.9** |
| Res101 [20] | 48 | 260 | 38.8 | Res101 [20] | 86 | 1029 | - | 44.9 |
| PVT-L [50] | 65 | 283 | 42.1 | Twins-B [9] | 89 | 1020 | 47.7 | 48.9 |
| TwinsP-L [9] | 65 | 283 | 46.4 | Swin-S [35] | 81 | 1038 | 47.6 | 49.5 |
| Swin-S [35] | 53 | 274 | 45.2 | Focal-T [61] | 85 | 1130 | 48.0 | 50.0 |
| Ortho-B | 53 | 297 | **49.0** | Ortho-B | 81 | 1057 | **49.8** | **51.2** |

Table 6: Ablation Study on three benchmarks using the Ortho-S backbone.

| | ImageNet-1K | | | COCO | | ADE20K |
|---|---|---|---|---|---|---|
| | #Param (M) | FLOPs (G) | Acc. (%) | $AP^b$ | $AP^m$ | mIoU |
| window sa | 24.0 | 4.5 | 82.6 | 46.2 | 42.0 | 47.5 |
| shifted window sa | 24.0 | 4.5 | 82.5 | 46.4 | 42.1 | 47.6 |
| dilated sa | 24.0 | 4.5 | 82.9 | 46.2 | 41.7 | 46.6 |
| ortho. sa | 24.0 | 4.6 | 83.2 | **47.0** | 42.3 | 48.0 |
| window/ortho. sa | 24.0 | 4.5 | **83.4** | **47.0** | **42.5** | **48.2** |
| no pos. | 23.4 | 4.4 | 82.2 | 44.0 | 40.4 | 44.6 |
| abs. pos. | 23.4 | 4.4 | 82.2 | 44.1 | 40.6 | 44.4 |
| rel. pos. | 23.4 | 4.4 | 82.4 | 43.4 | 40.1 | 44.6 |
| conv. pos. | 24.0 | 4.5 | **83.4** | **47.0** | **42.5** | **48.2** |
| w/o early convs | 23.9 | 4.2 | 82.6 | 46.5 | 41.9 | 47.1 |
| early convs | 24.0 | 4.5 | **83.4** | **47.0** | **42.5** | **48.2** |
| outside transition | 24.0 | 4.6 | 83.1 | 46.7 | 42.4 | 48.0 |
| inside transition | 24.0 | 4.5 | **83.4** | **47.0** | **42.5** | **48.2** |

## 4.3 Semantic Segmentation on ADE20K

**Experimental Settings.** Experiments on semantic segmentation are conducted on the ADE20K dataset[68]. We employ Semantic FPN [28] and Upernet [56] as the segmentation heads and replace the backbones with our Orthogonal Transformer. For a fair comparison, we follow the same setting of PVT [50] to train Semantic FPN 80k iterations and apply the setting of Swin [35] to train Upernet for 160k iterations. More details are provided in the Appendix.

**Results.** The results on semantic segmentation are listed in Table 5. Our method obtains large performance gain over other vision transformers. For methods that using Semantic FPN as the segmentation head, Orthogonal Transformer surpasses Twins by **+3.9**, **+2.6** on mIOU. The results with Upernet indicate that Orthogonal Transformer achieves **+2.9**, **+1.2** absolutely higher Multi-Scale(MS) mIOU than Focal Transformer, and reports **51.2** MS mIOU with 81M parameters. The comparisons on downstream tasks further validate the superiority of our method.

## 4.4 Ablation Study

In this section, we conduct ablation study to inspect the respective roles of each part in Orthogonal Transformer, using ImageNet-1K image classification, Mask R-CNN (1×) on COCO object detection, and Semantic FPN on ADE20K semantic segmentation.

**Self-Attention Mechanism.** Ablations of the orthogonal self-attention (SA) are reported in Table 6. Orthogonal SA outperforms window SA and dilated SA consistently on the three tasks. Since window

SA and dilated SA belong to fine-grained local SAs and coarse-grained global SAs respectively, the results may validate the superiority of orthogonal SA in capturing global dependency while preserving local details. The shifted window SA achieves similar performance with window SA. It is designed to allow for cross-window connection. In the Ortho-S backbone, DConv in MLP allows for cross-window interaction. That may be why shifted window SA is not superior to window SA herein. Besides, replacing half of orthogonal SA with window SA can lead to slight performance gain with 0.1G decrease of FLOPs, justifying the design of Orthogonal Transformer.

**Position Encoding.** Comparisons of different position embedding methods are reported in Table 6. Convolution position embedding (CPE) in PMLP performs better than those without position encoding and those with APE or RPE. The gains are specifically large on downstream high-resolution tasks, indicating that the design of our CPE is more suitable for arbitrary large input resolutions.

**Patch Embedding.** We borrow the idea of early convolutions from [57] to perform overlapped patch embedding. The results in Table 6 indicate that Ortho-S with early convolutions outperforms the counterpart with non-overlapped patch embedding.

**Transition.** Most of previous works using hierarchical architectures [35, 50] perform the transition of token number and feature dimension between stages. In this work, we adopt a different manner and employ the strided convolutions in PMLP to perform transition in the transformer block. The results in Table 6 imply that Ortho-S with inside transition can yield performance gain as well as computation reduction compared to the counterpart with outside transition.

## 5 Conclusion

We present a new vision transformer, called as Orthogonal Transformer, to serve as a strong general backbone for computer vision. The orthogonal self-attention mechanism is introduced to capture global information while exploring local correlations. Positional MLP is developed to handle position information within the transformer block for arbitrary input resolutions. Orthogonal Transformer achieves state-of-the-art performance in various vision tasks, including image classification, object detection, instance segmentation and semantic segmentation. We expect to apply Orthogonal Transformer for more vision tasks, such as image processing and video prediction.

## Acknowledgment

This work is partially funded by National Key Research and Development Program of China (Grant No. 2020AAA0140000), National Natural Science Foundation of China (Grant No. 62006228, U21B2045), and Youth Innovation Promotion Association CAS (Grant No. 2022132).

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
