# Orthogonal Transformer: An Efficient Vision Transformer Backbone with Token Orthogonalization

## A Proof of Theorem 1

Herein we provide the proof of Theorem 1 in the main text. Four lemmas with their proofs are given in advance, and the proof of Theorem 1 is in the last.

**Lemma A.1** *The Householder matrix* $\mathbf{H} = \mathbf{I} - \mathbf{2uu^T}$ *is symmetric and orthogonal, i.e.,* $\mathbf{H} = \mathbf{H^T}$ *and* $\mathbf{H}^{'} = \mathbf{H^T}$.

**Proof A.1** Since $\mathbf{H} = \mathbf{I} - \mathbf{2uu^T}, (\mathbf{uu^T})^T = \mathbf{uu^T}$, and $\mathbf{u^Tu} = 1$, we can conclude that

$$\mathbf{H^T} = (\mathbf{I} - \mathbf{2uu^T})^T = (\mathbf{I} - \mathbf{2uu^T}) = \mathbf{H}. \tag{1}$$

Therefore, $\mathbf{H}$ is symmetric.

$$\mathbf{HH^T} = (\mathbf{I} - \mathbf{2uu^T})(\mathbf{I} - \mathbf{2uu^T})^T = (\mathbf{I} - \mathbf{2uu^T})(\mathbf{I} - \mathbf{2uu^T}) = \mathbf{I} - \mathbf{4uu^T} + \mathbf{4uu^T} = \mathbf{I}. \tag{2}$$

Therefore, $\mathbf{H}$ is orthogonal.

**Lemma A.2** *Given any two non-zero vectors* $\mathbf{x}$ *and* $\mathbf{y}$ *with the same 2-norm, there exists a Householder transformation* $\mathbf{H}$ *satisfying* $\mathbf{Hx} = \mathbf{y}$.

**Proof A.2** We can construct the Householder matrix with vector $\mathbf{u} = \frac{(\mathbf{x}-\mathbf{y})}{||\mathbf{x}-\mathbf{y}||_2}$. Then,

$$\mathbf{H} = \mathbf{I} - \frac{2(\mathbf{x} - \mathbf{y})(\mathbf{x} - \mathbf{y})^\mathbf{T}}{||\mathbf{x} - \mathbf{y}||_\mathbf{2}^\mathbf{2}}, \tag{3}$$

and

$$\mathbf{Hx} = \mathbf{x} - \frac{2(\mathbf{x} - \mathbf{y})(\mathbf{x} - \mathbf{y})^\mathbf{T}}{||\mathbf{x} - \mathbf{y}||_\mathbf{2}^\mathbf{2}}\mathbf{x} = \mathbf{y}. \tag{4}$$

**Lemma A.3** *(QR factorization) A rectangular matrix* $\mathbf{A} \in \mathbb{R}^{n \times n}$ *can be factored into a product of an orthogonal matrix* $\mathbf{Q} \in \mathbb{R}^{n \times n}$ *and an upper triangular matrix* $\mathbf{R} \in \mathbb{R}^{n \times n}$ : $\mathbf{A} = \mathbf{QR}$, *where* $\mathbf{Q}$ *is the product of* $n - 1$ *orthogonal Householder matrices.*

**Proof A.3** The matrix $\mathbf{A}$ can be written as a block form $\mathbf{A} = [\mathbf{a_1}, \mathbf{a_2}, \cdots, \mathbf{a_n}]$, where $\mathbf{a_i} = [a_{i1}, a_{i2}, \cdots, a_{in}]^\mathbf{T}$. With Lemma A.2, the vector $\mathbf{a_1} = [a_{11}, a_{21}, \cdots, a_{n1}]^\mathbf{T}$ can be transformed to $[\times, 0, 0, \cdots, 0]^\mathbf{T}$ with a Householder transformation $\mathbf{H_1}$. We denote the non-zero value in the vector as "$\times$" for description convenience. This process can be formulated as:

$$\mathbf{H_1A} = \mathbf{H_1}[\mathbf{a_1}, \mathbf{a_2}, \cdots, \mathbf{a_n}] = \begin{bmatrix} \times & a_{12}^* & \cdots & a_{1n}^* \\ 0 & a_{22}^* & \cdots & a_{2n}^* \\ \vdots & \vdots & \ddots & \vdots \\ 0 & a_{n2}^* & \cdots & a_{nn}^* \end{bmatrix}. \tag{5}$$

36th Conference on Neural Information Processing Systems (NeurIPS 2022).

By repeating the above process to the first column of matrix $\begin{bmatrix} a_{22}^* & \cdots & a_{2n}^* \\ \vdots & \ddots & \vdots \\ a_{n2}^* & \cdots & a_{nn}^* \end{bmatrix}$ with a Householder transformation $\mathbf{H}_2^*$, the vector $[a_{22}^*, a_{32}^*, \cdots, a_{n2}^*]^{\mathbf{T}}$ can be transformed to $[\times, 0, 0, \cdots, 0]^{\mathbf{T}}$. Therefore, we conclude that

$$\mathbf{H_2 H_1 A} = \begin{bmatrix} 1 & 0 \\ 0 & \mathbf{H_2^*} \end{bmatrix} \mathbf{H_1 A} = \begin{bmatrix} \times & \times & a_{13}^{**} & \cdots & a_{1n}^{**} \\ 0 & \times & a_{23}^{**} & \cdots & a_{2n}^{**} \\ 0 & 0 & a_{33}^{**} & \cdots & a_{3n}^{**} \\ \vdots & \vdots & \ddots & \vdots \\ 0 & 0 & a_{n3}^{**} & \cdots & a_{nn}^{**} \end{bmatrix} \tag{6}$$

By repeating the above process for a total of $n-1$ times, we can obtain an upper triangular matrix:

$$\mathbf{H_{n-1} H_{n-2} \cdots H_1 A} = \begin{bmatrix} \times & \times & \times & \cdots & \times \\ 0 & \times & \times & \cdots & \times \\ 0 & 0 & \times & \cdots & \times \\ \vdots & \vdots & \vdots & \ddots & \vdots \\ 0 & 0 & 0 & \cdots & \times \end{bmatrix} = \mathbf{R}, \tag{7}$$

Considering that each Householder transformation $\mathbf{H_i}$ is orthogonal and symmetric, we conclude that

$$\mathbf{A} = \mathbf{H_1 H_2 \cdots H_{n-1} R} = \mathbf{QR} \tag{8}$$

**Lemma A.4** *If a $n \times n$ matrix $\mathbf{A}$ is not only upper triangular but also orthogonal, then $\mathbf{A}$ is a diagonal matrix.*

**Proof A.4** The $n \times n$ matrix $\mathbf{A}$ can be written as a block form $\mathbf{A} = \begin{bmatrix} \mathbf{A_1} & \mathbf{A_2} \\ \mathbf{0} & a_{nn} \end{bmatrix}$, where $\mathbf{A_1} \in \mathbb{R}^{(n-1) \times (n-1)}, \mathbf{A_2} \in \mathbb{R}^{(n-1) \times 1}$. Since $\mathbf{A}$ is orthogonal, we have

$$\mathbf{AA^T} = \begin{bmatrix} \mathbf{A_1} & \mathbf{A_2} \\ \mathbf{0} & a_{nn} \end{bmatrix} \begin{bmatrix} \mathbf{A_1^T} & \mathbf{0^T} \\ \mathbf{A_2^T} & a_{nn} \end{bmatrix} = \begin{bmatrix} \mathbf{A_1 A_1^T + A_2 A_2^T} & a_{nn} \mathbf{A_2} \\ a_{nn} \mathbf{A_2^T} & a_{nn}^2 \end{bmatrix} = \mathbf{I}. \tag{9}$$

Therefore, $a_{nn} = \pm 1$, $\mathbf{A_2} = \mathbf{0}$, and $\mathbf{A_1 A_1^T} = \mathbf{I} \in \mathbb{R}^{(n-1) \times (n-1)}$. The matrix block $\mathbf{A_1}$ is not only upper triangular but also orthogonal. By repeating the above process for a tota of $n$ times, we can conclude that $\mathbf{A}$ is diagonal with diagonal entries equal to $\pm 1$.

**Theorem A.5** *Every real orthogonal $n \times n$ matrix $\mathbf{A}$ is the product of at most $n$ real orthogonal Householder transformations.*

**Proof A.5** With Lemma A.3, we can upper triangularize the given real orthogonal matrix $\mathbf{A}$ as: $\mathbf{H_{n-1} H_{n-2} \cdots H_1 A} = \mathbf{R}$. Since $\mathbf{R}$ is both upper triangular and orthogonal as a product of orthogonal matrices, according to Lemma A.4, $\mathbf{R}$ is diagonal with diagonal entries equal to $\pm 1$. By constraining the entry of $\mathbf{R}$ to be positive when constructing the QR factorization in Lemma A.3, we have $r_{11} = r_{22} = \cdots = r_{n-1,n-1} = 1$.

If the last diagonal entry $r_{nn} = -1$, by setting $\mathbf{H_n} = \mathbf{I_n} - 2\mathbf{e_n e_n^T}$, we can obtain that

$$\mathbf{H_n H_{n-1} \cdots H_1 A} = \mathbf{H_n R} = \mathbf{I}. \tag{10}$$

As each Householder matrix $\mathbf{H_i}$ is its own inverse (Lemma A.1), we conclude that

$$\mathbf{A} = \mathbf{H_1 H_2 \cdots H_n}. \tag{11}$$

If the last diagonal entry $r_{nn} = 1$, then $\mathbf{R} = \mathbf{I}$. Because

$$\mathbf{H_{n-1} H_{n-2} \cdots H_1 A} = \mathbf{R}, \tag{12}$$

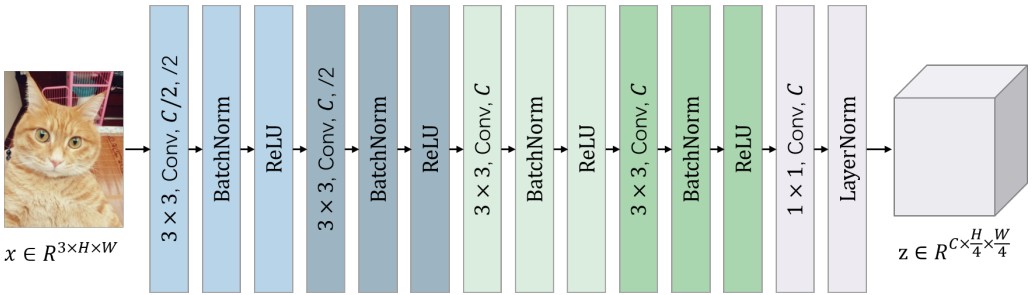

Figure I: Convolutional patch embedding in Orthogonal Transformer.

we can conclude that

$$\mathbf{A} = \mathbf{H_1 H_2 \cdots H_{n-1}}. \tag{13}$$

Thus, every real orthogonal $n \times n$ matrix $\mathbf{A}$ can be written as the product of at most $n$ Householder matrices.

## B  Experimental Settings

**ImageNet Image Classification.**    We follow the same training strategy in DeiT [15]. Specifically, all our models are trained from scratch for 300 epochs with the input size of $224 \times 224$. We use the AdamW optimizer with a cosine decay learning rate scheduler and 5 epoch linear warm-up. The initial learning rate, weight decay, and batch-size are 0.001, 0.05, and 1024, respectively. For Ortho-T, we use a smaller weight decay of 0.01. We adopt the strong data augmentation and regularization in [15], except for repeated augmentation [7] that does not improve performance. The augmentation settings are RandAugment [5] (randm9-mstd0.5-inc1) , Mixup [27] (prob = 0.8), CutMix [26] (prob = 1.0), Random Erasing [28] (prob = 0.25) and Exponential Moving Average [13] (ema-decay = 0.99996, we do not use it for Ortho-T), increasing stochastic depth [8] (prob = 0.1, 0.2, 0.4, 0.5 for Ortho-T, Orthon-S, Ortho-B and Ortho-L, respectively). For $384 \times 384$ input resolution, we fine-tune the models for 30 epochs with learning rate of 1e-5, weight decay of 1e-8 and bach-size of 512.

**COCO Object Detection and Instance Segmentation.**    We adopt Mask-RCNN [6] and Cascaded Mask R-CNN [1] based on MMDetection [2]. We train the models with two common settings: "1 ×" (12 training epochs) and "3 × +MS" (36 training epochs with multi-scale training). For the "1 ×" setting, images are resized to the shorter side of 800 pixels while the longer side is within 1333 pixels. The AdamW optimizer is used with learning rate of 0.0001, weight decay of 0.05 and batch-size of 16. The learning rate declines with decay rate of 0.1 at epoch 8 and epoch 11. For "3 × +MS", multi-scale input is used by resizing the shorter side of images between 480 and 800 pixels. The learning rate declines at epoch 27 and 33.

**ADE20K Semantic Segmentation.**    We adopt two popular semantic segmentation frameworks: Semantic FPN [10] and Upernet [21] based on MMSegmentation [4]. We apply Orthogonal Transformer pretrained on ImageNet-1K as the backbone network. For Semantic FPN, we follow the same setting of PVT [18] and train it for 80k iterations. For Upernet, we apply the setting in Swin [12] to train it for 160k iterations. All the models are trained on the input resolution of $512 \times 512$. The stochastic depth is the same as those used in ImageNet pre-training.

## C  Architecture Details

Fig. I and Fig.II show the detailed architectures of the convolutional patch embedding and the Positional MLP (PMLP) in Orthogonal Transformer.

**Convolutional Patch Embedding**    To perform overlapped patch embedding, we borrow early convolutions from [22] and apply 5 convolutions with the same setting of [22]. As shown in Fig. I,

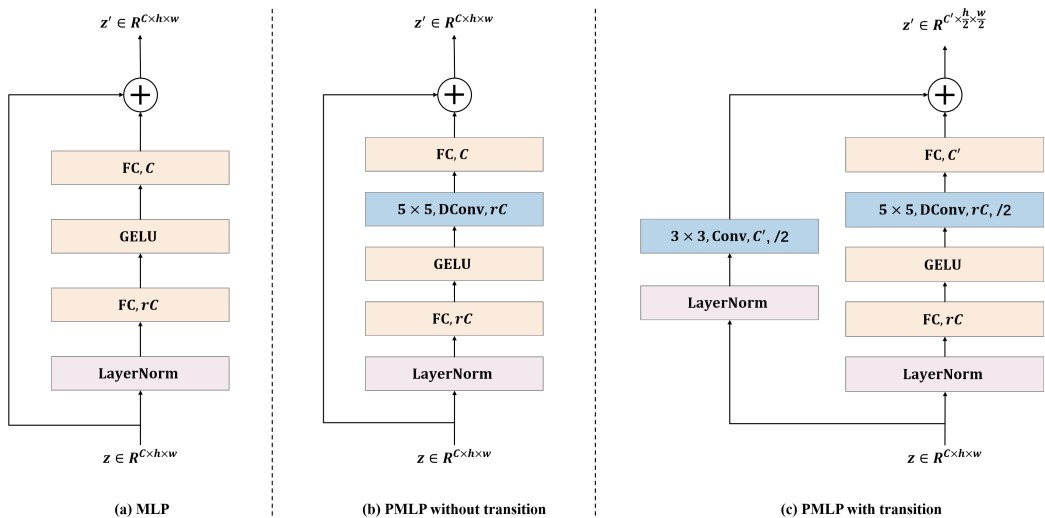

Figure II: Positional MLP (PMLP). (a) is the common MLP in transformer, (b) and (c) are the presented PMLP without and with transition, respectively. $r$ is the expansion ratio for MLP.

the first two convolutions are with the kernel-size of $3 \times 3$ and the stride of 2, while the next two are with the kernel-size of $3 \times 3$ and the stride of 2. BatchNorm and ReLU are utilized after each of them. The last convolution is with the kernel-size of $1 \times 1$, following by a LayerNorm layer.

**Positional MLP (PMLP).** Positional MLP is introduced to handle position information and two variants are developed according to performing transition or not. As shown in Fig. II, depth-wise convolution (DConv) with kernel-size of $5 \times 5$ is used after GELU in MLP. For PMLP with transition, we adopt a convolution with kernel-size of $3 \times 3$ and stride of 2 in the residual way and set the stride of DConv as 2. Spatial resolution is reduced to $\frac{h}{2} \times \frac{w}{2}$ and feature dimension is increased to $C'$.

Note that convolutional position embedding (CPE) has been applied in several previous works [3, 11]. Our PMLP differs from them in two manners. On the one hand, the location of DConv is carefully selected as next to GELU in MLP; on the other hand, PMLP with transition is developed and thereby transition is performed within transformer blocks. The ablation results in Table 6 in the main text have shown that transition inside is superior over outside. We inspect the location of DConv in Appendix D and demonstrate that DConv's location is crucial for the performance of CPE.

# D   Ablation Study

In this section, we make extra ablation studies to further demonstrate the crucial roles of the presented orthogonal self-attention (OSA) mechanism, the endogenous orthogonality construction and the design of Positional MLP (PMLP).

## D.1   Self-Attention Mechanisms

We compare our OSA with several previous efficient self-attention mechanisms, including linear SA [14], local SA in Swin [12], downsampled SA in PVT [18], dilated SA in GG [24]. For a fair comparison, we use Ortho-T as backbone and only vary the self-attention mechanism. Table I reports the results on three vision tasks, i.e., ImageNet image classification, COCO object detection and instance segmentation with Mask R-CNN (1× setting), and ADE20K semantic segmentation with Semantic FPN. Obviously, our OSA mechanism shows stronger performance than other efficient SA mechanisms for all the three tasks. Specifically, global SA mechanisms, including ours, linear SA, downsampled SA and dilated SA, outperform local SA mechanisms, implying that global modeling is critical for vision tasks. Our OSA outperforms other global SA mechanism for better capacity of local correlation learning.

Table I: Comparisons between different self-attention mechanisms using the Ortho-T backbone.

| Method | ImageNet-1K | | | COCO | | ADE20K |
| | #Param (M) | FLOPs (G) | Acc. (%) | $AP^b$ | $AP^m$ | mIoU |
|---|---|---|---|---|---|---|
| Linear [14] | 3.9 | 0.7 | 73.7 | 37.5 | 35.4 | 40.3 |
| Swin [12] | 3.9 | 0.7 | 73.1 | 37.9 | 35.7 | 39.9 |
| PVT [18] | 4.8 | 0.7 | 73.8 | 38.8 | 36.2 | 40.9 |
| GG [24] | 4.0 | 0.7 | 73.8 | 38.1 | 35.7 | 40.7 |
| Ours | 3.9 | 0.7 | **74.0** | **39.4** | **36.8** | **41.3** |

Table II: Comparisons on different backbones. Ours* employs a hierarchical structure like Swin.

| Backbone | Method | #Param (M) | FLOPs (G) | Acc. (%) |
|---|---|---|---|---|
| ViT-Ti | ViT | 5.7 | 1.3 | 72.2 |
| | Ours | 5.7 | 1.1 | 70.8 (-1.4) |
| | Ours* | 5.6 | 1.3 | 74.4 (+2.2) |
| Swin-T | Swin | 29 | 4.5 | 81.3 |
| | Ours | 28 | 4.5 | 81.6 (+0.3) |
| Ortho-S | Swin | 24 | 4.5 | 82.5 |
| | Ours | 24 | 4.5 | 83.4 (+0.9) |

**Comparison with linear transformer.** Here we give more discussions about linear transformer. Linear transformer [9, 17, 14] expresses self-attention as linear dot-product of kernel maps and reduce the complexity from quadratic to linear with respect to token number. Compared with linear self-attention, our OSA achieves better performance on the three tasks, indicating its stronger capacity in modeling global-local dependencies.

**Contribution Isolation.** To better isolate the contribution of OSA, we compare the self-attention mechanisms on different backbones, including vanilla ViT and vanilla Swin Transformer. The results are reported in Table II. For the vanilla ViT, directly replacing the standard SA with ours would reduce the computational cost but decrease the accuracy. Since the major superiority of the proposed orthogonal self-attention (OSA) over vanilla self-attention is enabling transformer to compute self-attention in high-resolution space with low computation complexity, adopting a large patch-size like ViT and computing self-attention in low-resolution space cannot validate the superiority of OSA. However, when merely using a hierarchical structure like Swin, our method achieves significant improvements over ViT under a similar cost. Compared with Swin, our method obtains consistent gains but the gain in the Ortho-S backbone is larger than that in the Swin-T backbone, implying that our presented structure would help make full use of OSA.

### D.2 Orthogonality

Orthogonality is the key aspect of the presented orthogonal self-attention. In this work, we construct an endogenously orthogonal matrix that is friendly to neural networks with gradient optimizers. Here we conduct experiments to verify the necessity of orthogonality and the superiority of the endogenous orthogonality construction. For clarity, we repeat the definition of OSA as:

$$f_{OSA}(Z) = \mathbf{A}^{\mathrm{T}} f_{MSA}(f_{LN}(\mathbf{A}Z)) + Z. \tag{14}$$

Apart from constructing $\mathbf{A} \in \mathbb{R}^{n \times n}$ explicitly with Householder matrices, orthogonality can be achieved by the following regularizer:

$$L_{rev} = \frac{1}{n^2} \|\mathbf{I} - \mathbf{A}^{\mathrm{T}}\mathbf{A}\|^2. \tag{15}$$

We compare our method with variants: $\mathbf{A}$ is randomly initialized and trained without $L_{rev}$; $\mathbf{A}$ is randomly initialized and trained with $L_{rev}$; $\mathbf{A}$ is orthogonally initialized and trained without $L_{rev}$; $\mathbf{A}$ is orthogonally initialized and trained with $L_{rev}$. Besides, two variants are added by replacing

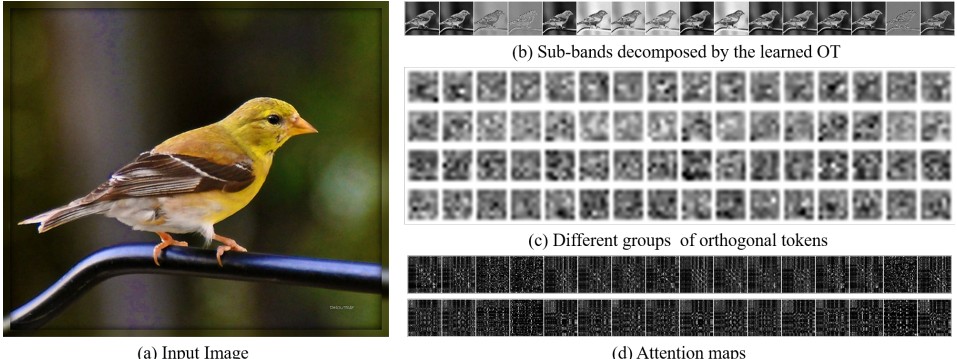

(a) Input Image

(b) Sub-bands decomposed by the learned OT

(c) Different groups of orthogonal tokens

(d) Attention maps

Figure III: Visualizations of orthogonal tokens and the attention maps. We take the last layer in the second stage as example and each column in the right corresponds to a specific orthogonal group, respectively. The image's sub-bands and the feature maps of the orthogonal tokens imply that the OT can capture different characteristics.

Table III: Orthogonality Analysis.

| replace $\mathbf{A}^{\mathrm{T}}$ with $\mathbf{B}^{\mathrm{T}}$ | Random Init. | Ortho. Init. | Ortho. Loss | presented Ortho. | Acc. (%) |
|---|---|---|---|---|---|
| | $\checkmark$ | | | | 73.5 |
| | $\checkmark$ | | $\checkmark$ | | 73.4 |
| | | $\checkmark$ | | | 73.8 |
| | | $\checkmark$ | $\checkmark$ | | 73.3 |
| $\checkmark$ | $\checkmark$ | | | | 69.3 |
| $\checkmark$ | $\checkmark$ | | $\checkmark$ | | 71.5 |
| | | $\checkmark$ | | $\checkmark$ | **74.0** |

$\mathbf{A}^{\mathrm{T}}$ with a different matrix $\mathbf{B}^{\mathrm{T}} \in \mathbb{R}^{n \times n}$: $\mathbf{A}$ and $\mathbf{B}$ are randomly initialized without additional regularization; $\mathbf{A}$ is randomly initialized, $\mathbf{B}$ is initialized as the pseudo inverse of $\mathbf{A}$ and they are regularized with the reverse regularization.

Table III shows that our method outperforms all the variants, implying the superiority of the endogenous orthogonality construction. Besides, under the same setting without orthogonal regularizer, the orthogonally initialized variant surpass the random initialized one. This demonstrates that orthogonality can provide a good initialization for transformation-based self-attention. The models with the orthogonal regularizer fail to achieve better performance than those without, implying that the orthogonal regularizer cannot directly enhance the performance and may need more efforts to tune good weight parameters (here the total loss is a simple sum of the classification loss and $L_{rev}$). Our orthogonality construction can maintain the orthogonality without extra orthogonal regularizer, avoiding hard tuning of hyper-parameters for it.

### D.3   Linear independency

One of the advantages of orthogonal transformations is that the tokens can be separated into linearly independent groups. Linear independency would help self-attention explore different properties of representation. As shown in Fig. III, the learned orthogonal transformation can split feature maps into groups that capture different characteristics. For example, some may capture low-frequency information, and some may capture high-frequency textures. The attention scores vary for different groups, leading to stronger capability of representation via different views.

In Table III, the last variant of replacing $\mathbf{A}^{\mathrm{T}}$ with $\mathbf{B}^{\mathrm{T}}$ with $L_{rev}$ can show the superiority of linear independency. Such variant remains the properties of reduced resolution, reversibility (the converged value of $L_{rev}$ is 1e-4 that is very close to zero), and token connections, but removes linear independency of orthogonal transformation. As shown in Table III, the performance degradation implies that linear independency is important for the overall performance.

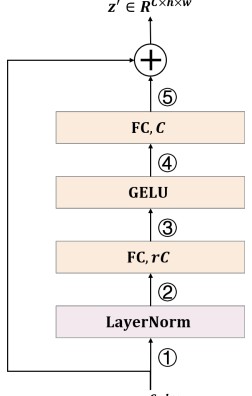

$z' \in R^{C \times h \times w}$

⑤

FC, $C$

④

GELU

③

FC, $rC$

②

LayerNorm

①

$z \in R^{C \times h \times w}$

Figure IV: MLP.

| | #Param (M) | FLOPs (G) | Acc. (%) |
|---|---|---|---|
| CPE [3] | 3.85 | 0.69 | 71.9 |
| CPE* [3] | 4.06 | 0.74 | 72.6 |
| PMLP-1 | 3.85 | 0.67 | 72.5 |
| PMLP-2 | 3.85 | 0.67 | 72.3 |
| PMLP-3 | 3.94 | 0.71 | 73.2 |
| PMLP-4 | 3.94 | 0.71 | **74.0** |
| PMLP-4* | 3.94 | 0.71 | 73.8 |
| PMLP-5 | 3.85 | 0.69 | 73.3 |

Table IV: Comparisons between different Convolutional Position Embeddings (CPEs). PMLP-n denotes PMLP's variant that applies the depth-wise convolution at the $n$-th location in Fig. IV. PMLP-4* is a variant with an additional GELU layer next to DConv.

Table V: Comparisons between different kernel sizes for the depth-wise convolution in PMLP.

| Kernel Size | #Param (M) | FLOPs (G) | Acc. (%) |
|---|---|---|---|
| $3 \times 3$ | 3.86 | 0.69 | 73.35 |
| $5 \times 5$ | 3.94 | 0.71 | 74.00 |
| $7 \times 7$ | 4.06 | 0.74 | **74.05** |

## D.4 Convolutional Position Embedding

We compare with CPE presented in [3] as well as several variants of our CPE with different settings. Results are reported in Table IV and Table V.

**Location of CPE** We conduct ablation study based on the Ortho-T backbone to validate the selected location of our CPE. We compare it with two variants of CPE used in [3]. One is originally used in [3] where the CPE module is put after the first transformer block in each stage, denoted as CPE in Table IV; the other is putting the CPE module before every transformer block, denoted as CPE* in Table IV. We also compare among several variants of our CPE by varying the locations of CPE in MLP. The possible locations are shown in Fig. IV. The adopted one in this work is PMLP-4, where the depth-wise convolution (DConv) is after GELU in MLP. We also compare with a variant with an additional GELU next to the inserted DConv, denoted as PMLP-4*.

Our adopted PMLP-4 surpasses the two variants of CPE [3] by a large margin. It also outperforms other variants with different DConv locations, including PMLP-1, PMLP-2, PMLP-3, PMLP-5. Its superiority over PMLP-1, PMLP-2 and PMLP-5 may come from that depth-wise convolution is performed in the feature space with larger dimension, i.e., $rC$ against $C$. PMLP-4 achieves better performance than PMLP-3 and PMLP-4*, which indicates that adding the non-linear activation GELU next to DConv may hurt the model performance.

**Kernel Size of DConv** We conduct ablation experiments to examine the effect of different kernel sizes for depth-wise convolution in PMLP. As shown in Table V, the kernel size of $5 \times 5$ obtains better performance than that of $3 \times 3$ by $+0.65\%$ accuracy, $+0.08$M model size and $+0.02$G FLOPs. The kernel size of $7 \times 7$ achieves similar performance with that of $5 \times 5$ by $+0.05\%$ accuracy, $+0.12$M model size and $+0.03$G FLOPs. We set the kernel size of DConv in Orthogonal Transformer as $5 \times 5$ to achieve a better trade-off between the performance and the model size as well as the computational complexity.

## D.5 Window Size for OSA and WSA

We conduct experiments to investigate the relationship between the window size and the performance. We vary the window size and build several variants on the backbone of Ortho-T. The results are reported in Table VI.

Table VI: The effect of different window sizes.

| $m_w$ for WSA | $m_o$ for OSA | Params (M) | FLOPs (G) | Acc. (%) |
|---------------|---------------|------------|-----------|----------|
| 4 | 8, 4, 2, 1 | 3.9 | 0.71 | 73.6 |
| 7 | 8, 4, 2, 1 | 3.9 | 0.71 | 74.0 |
| 14 | 8, 4, 2, 1 | 3.9 | 0.72 | 74.1 |
| 7 | 8, 4, 2, 1 | 3.9 | 0.71 | 74.0 |
| 7 | 4, 4, 2, 1 | 3.9 | 0.73 | 73.9 |
| 7 | 2, 2, 2, 1 | 3.9 | 0.86 | 74.1 |

The enlargement of the window size $m_w$ for WSA will bring about both performance gain and complexity increase. $m_w = 14$ can achieve slightly better performance than $m_w = 7$. We choose $m_w = 7$ following Swin Transformer.

For an input image of pixel-size $224 \times 224$, we set $m_o$ to ensure that the height/width is divisible by $m_o$ in every stage. Hence, the largest window sizes for OSA in the four stages are 8, 4, 2, 1, respectively. As shown above, when $m_o$ varies, the performances are close, but the complexity increases significantly when $m_o$ decreases. We set the values of $m_o$ as 8, 4, 2, 1 to achieve low computation complexity with competitive performance.

## E  Complexity Analysis

We provide an analysis of computational complexity and a comparison on running speed in the following.

**Computational Complexity.**  We rewrite the computational complexity of global self-attention (GSA), window self-attention (WSA), and our orthogonal self-attention (OSA) as:

$$\Omega(GSA) = 4hwC^2 + 2(hw)^2C, \tag{16}$$

$$\Omega(WSA) = 4hwC^2 + 2m_w^2hwC, \tag{17}$$

$$\Omega(OSA) = 4hwC^2 + \frac{1}{m_o^2}2(hw)^2C + 2m_o^2hwC, \tag{18}$$

where $m_w$ and $m_o$ are the window size in window partition of WSA and the orthogonal window size in OSA, respectively. When $m_o^2 \ll \sqrt{hw}$, the last term of $\Omega(OSA)$ can be ignored. When $m_o = \frac{\sqrt{hw}}{m_w}$, the second term of $\Omega(OSA)$ equals to the second term of $\Omega(WSA)$. We design the models of Orthogonal Transformer with $m_o = \frac{\sqrt{hw}}{m_w}$ to obtain a similar computational complexity with those using WSA. For example, as shown in Table 6 in the main text, our model achieves a similar FLOPs with its variant using WSA.

*Compared against dilated SA*. OSA has higher complexity than dilated self-attention but the gain is marginal, which is exactly the third term in Eq. (18), i.e., $2n_ohwC$. It can be ignored when $n_o \ll \sqrt{hw}$ (this is usually true for high-resolution vision tasks). Furthermore, OSA can achieve better performance than dilated self-attention. As shown in Table 6 in the main text, compared with dilated self-attention, the network with OSA has comparable FLOPs (4.6G vs 4.5G) and achieves better performance on ImageNet classification, COCO detection and ADE20K segmentation. Therefore, the marginal gain of complexity for OSA may be acceptable considering the obvious performance improvements.

**Running Speed.**  Since FLOPs cannot directly reflect the running speed of different models, we report the throughput on ImageNet image classification in Table V. The throughput is compared on a single A40 GPU with three input resolutions, including $224 \times 224$, $448 \times 448$ and $896 \times 896$.

Noticeably, for low-resolution inputs, our Orthogonal Transformer models are only faster than a few methods. This may be attributed to the extra computation costs introduced by the orthogonal transformation and the DConv in PMLP. The running speed for different operations depends on

Figure V: Model throughput comparison on different resolutions.

| Model | Acc. (%) | FLOPs (G) | $224 \times 224$ | $448 \times 448$ | $896 \times 896$ |
|---|---|---|---|---|---|
| DeiT-Ti [15] | 72.2 | 1.3 | 3216 | 507 | 50 |
| PVTv2-b0 [19] | 70.5 | 0.6 | 2397 | 525 | 79 |
| T2T-7 [25] | 71.7 | 1.1 | 2314 | 444 | 53 |
| Ortho-T | 74.0 | 0.7 | 2259 | 531 | 90 |
| DeiT-S [15] | 79.9 | 4.6 | 1528 | 244 | 24 |
| PVT-S [18] | 79.8 | 3.8 | 1029 | 230 | 37 |
| T2T-14 [25] | 80.7 | 5.2 | 1120 | 190 | 20 |
| CVT-13 [20] | 81.6 | 4.5 | 1085 | 174 | 15 |
| Swin-T [12] | 81.3 | 4.5 | 919 | 231 | 58 |
| CaiT-XS24 [16] | 81.8 | 5.4 | 584 | 44 | 3 |
| Focal-T [23] | 82.2 | 4.9 | 327 | 83 | 21 |
| Ortho-S | 83.4 | 4.5 | 632 | 148 | 26 |
| PVT-L [18] | 81.7 | 9.8 | 510 | 113 | 18 |
| T2T-24 [25] | 82.2 | 13.2 | 570 | 92 | 9 |
| CVT-21 [20] | 82.5 | 7.1 | 714 | 116 | 11 |
| Swin-S [12] | 83.0 | 8.7 | 565 | 142 | 36 |
| CaiT-S24 [16] | 82.7 | 9.4 | 447 | 31 | 2 |
| Focal-S [23] | 83.5 | 9.1 | 202 | 52 | 13 |
| Ortho-B | 84.0 | 8.6 | 432 | 102 | 19 |
| DeiT-B [15] | 81.8 | 17.5 | 664 | 110 | 11 |
| Swin-B [12] | 83.3 | 15.4 | 415 | 105 | 26 |
| CaiT-S48 [16] | 83.5 | 18.6 | 146 | 8 | 0.5 |
| Focal-B [23] | 83.8 | 16.0 | 153 | 39 | 10 |
| Ortho-L | 84.2 | 15.4 | 277 | 64 | 11 |

bottom implementations in deep learning framework (Pytorch is used in this work), resulting in speed gaps even under the similar complexity of FLOPs. However, when the resolution increases, our models become competitive and run faster than more methods, especially for those using global self-attention, such as DeiT [15], PVT [18], T2T [25], CVT [20] and CaiT [16]. Remarkably, our models are consistently faster than the previous state-of-the-art Focal Transformer [23] for the three resolutions. Moreover, our models achieve better accuracy performance and have surpassed those with faster speed (such as Swin Transformer [12]) by a wide margin.

Above all, the major motivation for the presented orthogonal self-attention that captures global dependency while preserving local details is to reduce the computation cost of self-attention for high-resolution vision tasks, such as object detection, instance segmentation, and semantic segmentation. The strong performance and the speed superiority for high-resolution inputs validate the success of Orthogonal Transformer on both effectiveness and efficiency.

## F    Limitations and Broader Impacts

While our proposed Orthogonal Transformer achieves superior performance on various vision tasks, the orthogonal transformation in orthogonal self-attention and depth-wise convolution in Positional MLP introduce extra computational cost. Though competitive or better with high-resolution inputs, the running speed of our models is slower than some of their counterparts with low-resolution inputs. However, the great performance improvement brought by our efficient global self-attention and novel architecture design may compensate such a limitation in some degree.

Another limitation of our work may be that we didn't take experiments to apply our method to unsupervised learning, video or NLP tasks due to limited efforts and computational resource. We would like to explore such possibilities in future work.

This work is a purely academic study and we are not aware of any direct negative social impact in our work. Possible malicious use of our models is a problem that can be faced by the entire field. Related discussions are beyond the scope of our research.