# OpenReview forum: "Orthogonal Transformer: An Efficient Vision Transformer Backbone with Token Orthogonalization"
_NeurIPS.cc/2022/Conference — NeurIPS 2022 Accept_

### Official Review · Reviewer_SBMR · 2022-07-09

**Rating:** 6
**Confidence:** 4
**Soundness:** 3 good
**Presentation:** 3 good
**Contribution:** 3 good

**Summary:**

The paper proposed an token orthogonalization layer, there self-attention is performed on groups of orthogonalized tokens. The paper then designs the Orthogonal Transformer (OT) which combines the orthogonal self-attention layer with a pyramid Transformer architecture, positional MLPs (adding a depthwise-convolution into the MLP layer), early convolutions, and a novel downsampling method between stages. OT performs well on ImageNet-from-scratch, COCO detection and instance segmentation, and ADE20k semantic segmentation.

**Questions:**

Please address my main concern about the orthogonalization layer. Given that the orthogonal self-attention layer is marketed the main contribution of the paper it would be useful to have an experiment that demonstrates the value of this layer in isolation. I think the best way to do this would be to run a vanilla ViT/DINO setup with just the OT layer replacing self-attention, but without any of the additional changes/tricks. This would help determine with the OT layer is useful in isolation, or it requires the other network modifications in addition to be a useful component.

Second, you answered "Yes" to "Did you include the code, data, and instructions needed to reproduce the main experimental result?". I did not see any mention of opensourced code, will code be made available?

**Limitations:**

Section F in the appendix dedicated to limitations and societal impact. The societal impact is adequately addressed. The limitations section is brief, but I think it is adequate, and i believe captures the main limitation that the study is restricted only to image-based tasks.

**Strengths And Weaknesses:**

Strengths

* I like the proposal of constructing a trainable orthogonalization module from the product of Householder matrices. The orthogonal self-attention layer is fairly simple and general, and I think is a nice contribution.

* The experiment cover multiple tasks, appear thorough, and compare to many state-of-the-art alternatives.

* There is an ablation of the four components of the orthogonal Transformer.

Weaknesses

* I think the main weakness is the large number of moving parts in OT: the combination of both windowed attention and orthogonal attention, addition convolutions at the start and in the middle of the network, and a new downsampling mechanism. The combination of these factors significantly increases the complexity of the network over the original Transformer/Vision Transformer, potentially limiting adoption. The ablation study shows that some of the "minor" components have an equal-or-greater impact than the orthogonal attention layer which is the main "selling point" for OT. E.g. Conv position embeddings improve the scores on all tasks over absolute position embeddings more than orthogonal attention. Therefore, it feels like the orthogonal self-attention layer is not the key driver of performance in OT.

* Given that the network contains many vision-specific components (convolutions), I feel that "Orthogonal Transformer" is over-selling or over-generalizing the network. A name like "Orthogonal Vision/Image Transformer" would make it more clear that this is a vision-specific variant.

* There are a few typos, such as:

"tokens has a lower resolution" -> have

"singe OTB" -> single OTB

---

> ### Author Response · Authors · 2022-08-02
> **On the orthogonalization layer and name**
>
> We sincerely appreciate your hard work and positive comments on our paper. We will address your concerns in the following parts.
>
> **Q1:** About the orthogonalization layer.
>
> **R1**: Thanks for your constructive suggestion. It is a nice idea to run a vanilla ViT/DINO setup with just the OT layer replacing self-attention. However, vanilla ViT uses a large patch-size (e.g., 16 in ViT-S) to reduce the number of tokens. The major superiority of the proposed orthogonal self-attention (OSA) over vanilla self-attention is enabling transformer to compute self-attention in high-resolution space with low computation complexity. Adopting a large patch-size and computing self-attention in low-resolution space cannot validate the superiority of OSA. If we set a small patch-size for ViT, the complexity would increase explosively. For example, the value of FLOPs increases from 4.6G to 157G when the patch-size changes from 16 to 4 (which is the commonly used patch size in many vision transformers). Limited by the computing resources, we choose to compare our method with Swin Transformer rather than vanilla ViT in a simple backbone.
>
> Specifically, we adopt the setup of Ortho-T except that we replace convolutional position encoding with absolute position encoding, adopt the same patch embedding with vanilla ViT, and employ outside transition like Swin Transformer. We compare the proposed orthogonal self-attention layer with Swin’s self-attention layer. The proposed OSA achieves better performance than shifted window attention, specifically 69.9% vs 67.6% on ImageNet image classification. This verifies the effectiveness of OSA.
>
> Besides, we also conduct comparisons between different self-attention mechanisms in Appendix D.1. Except the self-attention layer, other parts in the transformer network are kept same. Our OSA outperforms other self-attention mechanisms consistently on three vision tasks, i.e., ImageNet Image classification, COCO object detection and instance segmentation, and ADE20K semantic segmentation. This also verifies that the orthogonal self-attention mechanism can boost the performance on various vision tasks.
>
> **Q2:** Name.
>
> **R2:** Thanks for this constructive suggestion. We follow previous vision transformer works, such as Swin Transformer and Focal Transformer, to name our model as Orthogonal Transformer. We expect to generalize it to other fields, like NLP and video processing in the future work (possible manner may be replacing the 2D convolutions with 1D/3D ones). We will modify the name if accepted.
>
> **Q3:** Typos.
>
> **R3** Thanks. We will correct them and check the paper carefully.
>
> **Q4:** Code.
>
> **R4:** Thanks for your kind reminder. We will release the code as soon as possible if accepted.

---

### Official Review · Reviewer_hQkC · 2022-07-11

**Rating:** 5
**Confidence:** 4
**Soundness:** 2 fair
**Presentation:** 3 good
**Contribution:** 2 fair

**Summary:**

This paper proposed an efficient Transformer backbone, Orthogonal Transformer, for efficient processing of visual inputs. The backbone is mainly built on the proposed efficient self-attention mechanism, orthogonal self-attention (OSA), to model dependencies between tokens in the orthogonal space. The proposed backbone achieves excellent performance-FLOPs tradeoff on visual tasks.

**Questions:**

The questions about the paper are listed in the Weaknesses.

**Limitations:**

Yes.

**Strengths And Weaknesses:**

### Strengths
The performances of the proposed backbone are superior.
The paper is well written and easy to follow, and the illustration is clear to me.

### Weaknesses
The motivation is not clear to me. Why do other efficient self-attention approaches fail? The author claimed other efficient self-attention mechanisms "lose fine-level details for coarse global self-attention or hurting long-range modeling for local self-attention" (L35-36). However, I cannot find any pieces of evidence to support the claim. In addition, how does the proposed orthogonal self-attention (OSA) work? The author claimed OSA "captures global dependency without losing fine-level details" (L37-38). I have found no evidence to support the claim, either theoretically or empirically.

The contribution is not significant. I believe the only novel part of this paper is the proposed efficient self-attention approach.  The proposed Positional MLP in Sec 3.4 has been studied in PVT v2 [CVMJ 22], CeiT [ICCV 21], and CMT [CVPR 22]. So I would not say this is the novel part.

The ablation is not sufficient. In Tab 6, the authors compared the proposed OSA with window SA and dilated SA. How about the window SA with shified window like Swin?

---

> ### Author Response · Authors · 2022-08-02
> **On motivation, contribution and ablation**
>
> We thank the reviewer for recognizing the positive aspects of our paper, and we will address the reviewer’s concerns in the following parts.
>
> **Q1:** Motivation.
>
> **R1:** Thanks. We would like to clarify that the claim *"lose fine-level details for coarse global self-attention or hurting long-range modeling for local self-attention"* is borrowed from and proved in [Tang et al., 2022], [Yu et al., 2021]. Related descriptions are listed below.
>
> *To reduce the computational cost, the PVT uses downsampled keys and values, which is harmful to capture pixel-level details. In comparison, the Swin Transformer restricts the attention in local windows in a single attention block, which might hurt long range dependencies* (Tang et al., 2022)
>
> *Spatial-reduction attention can reduce memory and computation costs to learn high-resolution feature maps, yet with a price of losing details which are expected from the high-resolution feature maps. Adopting self-attention within local windows is efficient with linear complexity, but it sacrifices the most significant advantage of Transformers in modeling long-range dependencies.* (Yu et al., 2021)
>
> *Although the Glance branch can effectively capture long-range representations, it misses the local connections across partitions* (Yu et al., 2021)(Glance branch uses dilated self-attention)
>
> The proposed OSA can capture global dependency without losing fine-level details for two reasons: 1) It can capture global dependency because it can always cover the whole images; 2) Orthogonal transformation is reversible and builds local connections among adjacent tokens explicitly, leading to strong capacity of local correlation learning.
>
> As shown in Table 6 and Table I (in Appendix), compared with other self-attention (SA) mechanisms, orthogonal self-attention performs better on various vision tasks, which may be attributed to its strong capacity of learning global and local dependencies. Besides, from the visualization of orthogonal tokens, we found that some of them capture low-frequency global information while some capture high-frequency local textures. The attention scores illustrate that OSA can capture global dependency on low-frequency tokens and model local correlation on high-frequency tokens. The comparisons on the attention scores against other SA mechanisms also show the superiority of OSA in modeling global and local dependencies.  We will add visual comparisons on tokens and attention scores if accepted.
>
> [1] Shitao Tang, Jiahui Zhang, Siyu Zhu, and Ping Tan. Quadtree attention for vision transformers. ICLR, 2022.
>
> [2] Qihang Yu, Yingda Xia, Yutong Bai, Yongyi Lu, Alan L Yuille, and Wei Shen. Glance-and-gaze vision transformer. NeurIPS, 2021.
>
> **Q2:** Contribution.
>
> **R1:** Thanks. We would like to highlight that the key novelty is that we construct a trainable orthogonalization module and the orthogonal self-attention layer is simple and general. The orthogonality is realized by constructing an endogenously orthogonal matrix that can be optimized as arbitrary orthogonal matrices without extra regularizer. Based on it, Orthogonal Transformer is built and achieves competitive performance on various vision tasks. We believe that Orthogonal Transformer can serve as a strong general baseline and considerably advance a broad range of vision tasks.
>
> Besides, we agree that depth-wise convolution (DConv) has been applied in previous works. Our major difference is that strided DConv is used to perform downsampling within the transformer block. In this way, we do not need extra patch merging layer to perform feature downsampling. Table 6 shows that it can achieve better performance than outside the transformer block. We also conduct extensive experiments to investigate the best location of DConv in Appendix D.3. Note that there exist various manners to employ DConv in vision transformers (e.g., PVT v2, CeiT, CMT and ours adopt DConv in different locations). We believe our work is suggestive to find appropriate manner of incorporating depth-wise convolution into transformers. We will discuss about PVT v2, CeiT, and CMT along with our differences if accepted.
>
> **Q3:** Shifted WSA.
>
> **R3:** Thanks. We have conducted experiments with variants using shifted window. As reported below, shifted window SA achieves similar performance with window SA. The shifted windowing scheme is designed to allow for cross-window connection. In the Ortho-S backbone, DConv in MLP allows for cross-window interaction. That may be why shifted window SA is not superior to window SA herein.
>
> | Method | #Param (M) | FLOPS (G) | Acc. (\%) | AP$^b$| AP$^m$|mIoU|
> | :------ | -------: |-------: |-------: |-------: |-------: |-------: |
> |window sa | 24.0 | 4.5 | 82.6 | 46.2 | 42.0 | 47.5|
> |shifted window sa | 24.0 | 4.5 | 82.5 | 46.4 | 42.1 | 47.6|
> |dilated sa| 24.0 | 4.5 | 82.9 | 46.2 | 41.7 | 46.6|
> |ortho. sa| 24.0 | 4.6 | 83.2 | 47.0 | 42.3 | 48.0|
> |window/ortho. sa | 24.0 | 4.5 | 83.4 | 47.0 | 42.5 |48.2 |

---

### Official Review · Reviewer_2jJ5 · 2022-07-11

**Rating:** 5
**Confidence:** 5
**Soundness:** 3 good
**Presentation:** 3 good
**Contribution:** 3 good

**Summary:**

The paper proposes a new efficient self-attention form  (Orthogonal Self-Attention, OSA) that conducts self-attention within groups, where orthogonalization of window tokens are performed first before forming the groups. The final architecture consists of alternative OSA and window self-attention (WSA) as attention, with FFN equipped with depth-wise convolutions. It shows competitive results on ImageNet classification, COCO object detection, ADE20k semantic segmentation.

**Questions:**

1.	As mentioned in weakness, The reversible operation is the novel part, but no ablation on that. Experiments do not show the essentialness of using orthogonal matrix instead of regular one. What if we keep the form (i.e. using non-orthogonal matrix to transform tokens within local windows, and then group them in same way as current strategy to perform self-attention), then no token reverse as inverse of weights is hard to get, maybe replacing multiplying the inverse of A with another new weights A’.

2.	Line 41 mentioned that tokens have lower resolution in the orthogonal space, can the authors provide more details regarding this? From my understanding, the new tokens obtained from multiplying the orthogonal matrix will not change dimensions, Z’ = AZ, Z’ is (n x c), A is (n x n), Z is (n x c), n = h x w.  The computation saving for attention here is from conducting group-wise attention, which does not seem to be brought from orthogonalization of tokens, but from window and group partitions of tokens.
3.	How to choose window size for WSA and OTB? Will different size affect the final performance?



-----post-rebuttal---
Thanks for the authors presenting the ablation study results on the essentialness of orthogonalization during rebuttal. Still, considering that the gain from ortho. sa is limited, 0.3% compared to dilated sa as in Table 6, I'll keep my initial rating as borderline accept.

**Strengths And Weaknesses:**

Strengths:
 	The way to construct orthogonal matrix endogenously by multiplying householder reflectors is novel and interesting. The OSA form proposed enabled the token connections within local window and links across windows.
	The experiments results show competitive performance across different vision tasks, including image classification, object detection, semantic segmentation.

Weaknesses:
	The essentialness of using orthogonal matrix is not studied.
The whole OSA process
1.	connects tokens within local windows with local window token orthogonalization, is serves as MLP layer within local windows, except the weight matrix of MLP is naturally orthogonal.
2.	Connects tokens beyond local windows by forming new groups across previous local window.
3.	Token reverse as the inverse of orthogonal matrix is easy to get, just the transpose of the matrix.
            Step 2 can be done regardless of the weight matrix of this local window MLP is orthogonal or not.
	Step 3 is the vital part that only orthogonal matrix weight can perform, I believe this should be studied, which is not presented, for validating the essentialness of using orthogonal matrix rather than just following the form that connects local and connects beyond local windows.

---

> ### Author Response · Authors · 2022-08-02
> **On essentialness of orthogonality, resolution, window size for WSA and OTB**
>
> We thank the reviewer for recognizing the contributions of our paper and giving us constructive suggestions. We will address the reviewer’s concerns in the following parts.
>
> **Q1:** The essentialness of using orthogonal matrix.
>
> **R1:** Thank you for your constructive suggestion. We have investigated the essentialness of orthogonality in Appendix D.2. We compare our method with variants where $A$ is initialized randomly or orthogonally, and $A$ is trained with or without the orthogonal loss $L_{ortho} = \frac{1}{n^2}\| \mathbf{I} - \mathbf{A}^{\mathrm{T}}\mathbf{A}\|^2$ or not regularized. When trained without $L_{ortho}$, the orthogonality of $A$ is not enforced.
> As shown in Table II (repeated below), the performance degradation without orthogonal regularization implies the essentialness of orthogonality. More details are in Appendix D.2.
>
> |Random Init.| Ortho. Init. | Ortho. Loss | presented Ortho. | Acc. (\%)|
> |:-----------|----------:|----------:|----------:|----------:|
> |$\surd$ | | |  | 73.5 |
> |$\surd$ | | $\surd$ |  | 73.4 |
> ||$\surd$  | |  | 73.8 |
> ||$\surd$ | $\surd$ |  | 73.3 |
> ||$\surd$  | | $\surd$ |  **74.0**  |
>
> Thanks for your precious suggestion. We realized the above analysis neglects the case where the matrices are different in Step 1 and Step 3. We further conduct experiments using two different matrices, i.e., $\mathbf{A}\in \mathbb{R}^{n\times n}$ in Step 1 and $\mathbf{B}\in \mathbb{R}^{n\times n}$ in Step 3. We explore two variants: $\mathbf{A}$ and $\mathbf{B}$ are randomly initialized without additional regularization; $\mathbf{A}$ is randomly initialized, $\mathbf{B}$ is initialized as the pseudo inverse of $\mathbf{A}$ and they are regularized with the reverse loss $L_{rev} = \frac{1}{n^2}\| \mathbf{I} - \mathbf{B}\mathbf{A}\|^2$. As shown below, our method using orthogonal matrix surpasses those without orthogonal matrix, validating the essentialness of orthogonality. The reversibility between Step 1 and Step 3 can improve the performance, but the regularization way with $L_{rev}$ lags the presented endogenously orthogonality construction.
> |Reverse Loss |presented Ortho. | Acc. (\%)|
> |:----------|------------:|------------:|
> |||69.3|
> |$\surd$||71.5|
> ||$\surd$|74.0|
>
>
> **Q2:** Lower resolution in the orthogonal space.
>
> **R2:** Thanks. It is true that the tokens obtained from multiplying the orthogonal matrix will not change the total dimensions. Line 41 means that tokens in the orthogonal space have lower spatial resolution. As described from Line 157 to Line 163, token orthogonalization would transform the input feature $Z\in \mathbf{R}^{(h\times w)\times c}$ into $n_o$  groups of orthogonal tokens $\hat{Z}^j\in \mathbf{R}^{(\frac{h}{m_o}\times \frac{w}{m_o}) \times c}$ (where $j=0,\ldots,n_o-1$). The spatial dimension is reduced from $h \times w$ to $\frac{h}{m_o}\times \frac{w}{m_o}$. We will clarify this in the final version.
>
> **Q3:** Window size for WSA and OTB.
>
> **R3:** Following Swin Transformer, we set the window size for WSA as 7. We set the window size for OSA to make the complexity of OSA similar with that of WSA in the same stage. For clarity, we repeat the complexities of OSA and WSA in the following:
> \begin{equation}
>   \Omega(OSA) = 4hwC^2 + \frac{1}{m_o^2}2(hw)^2C + 2n_ohwC,
> \end{equation}
> \begin{equation}
>     \Omega(WSA) = 4hwC^2+2m_w^2hwC.
> \end{equation}
> The window size $m_o$ for OSA is set as $m_o=\frac{\sqrt{hw}}{m_w}$, leading to the second terms in $\Omega(OSA)$ and $ \Omega(WSA) $ equaling to each other. For example, given an input image of pixel-size $224\times 224$, for the first stage where $h=w=56$, we set $m_w=7$ and $m_o=8$.
>
> Thanks for your constructive suggestion. We conduct additional experiments to investigate the relationship between the window size and the performance. We vary the window size and build several variants on the backbone of Ortho-T. The results are reported in the following.
> |$m_w$ for WSA |$m_o$ for OSA| Params (M) | FLOPs (G)| Acc. (\%)|
> |----------:|----------:|----------:|----------:|----------:|
> Varying $m_w$
> |4 | 8,4,2,1 | 3.9| 0.71 | 73.6|
> |7 | 8,4,2,1 | 3.9| 0.71 | 74.0|
> |14|8,4,2,1 | 3.9| 0.72 | 74.1|
> Varying $m_o$
> |7 | 8,4,2,1 | 3.9| 0.71 | 74.0|
> |7 | 4,4,2,1 | 3.9| 0.73 | 73.9|
> |7 | 2,2,2,1 | 3.9| 0.86 | 74.1|
>
> The enlargement of the window size $m_w$ for WSA will bring about both performance gain and complexity increase. $m_w=14$ can achieve slightly better performance than $m_w=7$. We choose $m_w=7$ following Swin Transformer.
>
> For an input image of pixel-size $224\times 224$, we set $m_o$ to ensure that the height/width is divisible by $m_o$ in every stage. Hence, the largest window sizes for OSA in the four stages are 8, 4, 2, 1, respectively. As shown above, when $m_o$ varies, the performances are close, but the complexity increases significantly when $m_o$ decreases. We set the values of $m_o$ as 8, 4, 2, 1 to achieve low computation complexity with competitive performance.

---

### Official Review · Reviewer_GmQX · 2022-07-11

**Rating:** 4
**Confidence:** 4
**Soundness:** 3 good
**Presentation:** 3 good
**Contribution:** 2 fair

**Summary:**

This paper introduces orthogonal transformation of tokens and combines it with the idea of interleaving window self attention (WSA) and dilated self-attention (DSA) to capture both local and global interactions without incurring the prohibitive cost of full global self attention. They showed that initialization and optimization of orthogonal matrices can be simplified by using Houholder transformations and leveraging already established optimization techniques to learn these matrices. In addition, the paper also make use of Positional MLPs where MLPs are equipped with depth-wise convolutions to allow for downsampling within the transformer block.

Authors individually ablated different techniques proposed in the paper and combined the best configurations to achieve top performance on wide range of image tasks.

**Questions:**

line 44: Authors claim that one of the advantages of orthogonal transformations is that the tokens can be separated into linearly independent groups. But, there is no further explanation as to why this linear independency is important or how this contributes to the overall performance of the model.

equation 5: Authors present complexity analysis in comparison with the global self-attention. But, it is only fair to compare against dilated self attention complexity.

**Limitations:**

Given that the paper is just concatenation of already existing ideas with minimal novelty of newly introduced techniques, I am not inclined to accept the paper.

**Strengths And Weaknesses:**

The main differentiating contribution of the paper is using the Householder transformation trick to introduce orthogonal transformations. This idea is coupled with already existing techniques of window self-attention and dilated self-attention. Authors present extensive experiments ablating each individual technique introduced. The paper is also well written and easy to read and understand.

---

> ### Author Response · Authors · 2022-08-02
> **On linear independency, complexity analysis, and novelty**
>
> We thank the reviewer for recognizing the positive aspects of our paper, and we will address the reviewer’s concerns in the following parts.
>
> **Q1:**  Linear independency.
>
> **R1:** Thank you for your constructive suggestion. Linear independency would help self-attention explore different properties of representation. We empirically found that the learned orthogonal transformation can split feature maps into groups that capture different characteristics. For example, some may capture low-frequency information, and some may capture high-frequency textures. The attention scores vary for different groups, leading to stronger capability of representation via different views. If accepted, we will add visualizations of orthogonal tokens as well as the attention scores to show how linear independency help explore image properties in different views.
>
> Besides, we also conduct ablation experiments to validate the role of linear independency. We modify OSA
> \begin{equation}
> f_{OSA}(Z) = \mathbf{A}^{\mathrm{T}} f_{MSA}(f_{LN}( \mathbf{A}Z)) + Z ,
> \end{equation}
> as
> \begin{equation}
> f_{OSA}(Z) = \mathbf{B} f_{MSA}(f_{LN}( \mathbf{A}Z)) + Z ,
> \end{equation}
> where $\mathbf{A}\in \mathbb{R}^{n\times n}$ is randomly initialized and $\mathbf{B}\in \mathbb{R}^{n\times n}$ is initialized as the pseudo inverse of $A$. They are regularized with
> \begin{equation}
> L_{rev} = \frac{1}{n^2}\| \mathbf{I} - \mathbf{B}\mathbf{A}\|^2.
> \end{equation}
> Such variant remains the properties of reduced resolution, reversibility (the converged value of $L_{rev}$ is 1e-4 that is very close to zero), and token connections, but removes linear independency of orthogonal transformation. As shown in the following, the performance degradation implies that linear independency is important for the overall performance.
> | Method | Acc. (\%) |
> | :---------- |  -------: |
> | w/o Linear Independency| 71.5 |
> | with Linear Independency | 74.0 |
>
> **Q2:** Complexity analysis.
>
> **R2:** Thanks. We compare complexity against the global self-attention to show that the proposed OSA can reduce the computation complexity of self-attention, especially for high-resolution vision tasks.
>
> OSA has higher complexity than dilated self-attention but the gain is marginal, which is  exactly the third term in Eq. (5), i.e., $2n_o hwC$. It can be ignored when $n_o\ll\sqrt{hw}$ (this is usually true for high-resolution vision tasks).
> Furthermore, OSA can achieve better performance than dilated self-attention.
> As shown in Table 6 (the related parts are repeated in the following), compared with dilated self-attention, the network with OSA has comparable FLOPs (4.6G vs 4.5G) and achieves better performance on ImageNet classification, COCO detection and ADE20K segmentation.
> Therefore, we believe the marginal gain of complexity for OSA is acceptable considering the obvious performance improvements.
>
> We will compare the complexity against dilated self-attention with a detailed analysis if accepted.
>
> | Method | #Param (M) | FLOPS (G) | Acc. (\%) | AP$^b$| AP$^m$|mIoU|
> | :------ | -------: |-------: |-------: |-------: |-------: |-------: |
> |dilated sa| 24.0 | 4.5 | 82.9 | 46.2 | 41.7 | 46.6|
> |ortho. sa| 24.0 | 4.6 | 83.2 | 47.0 | 42.3 | 48.0|
>
> **Q3:** Novelty.
>
> **R3:** Thanks. We would like to highlight that the key novelty is that we construct a trainable orthogonalization module and the orthogonal self-attention layer is simple and general. The orthogonality is realized by constructing an endogenously orthogonal matrix that can be optimized as arbitrary orthogonal matrices without extra regularizer. Based on it, Orthogonal Transformer is built and achieves competitive performance on various vision tasks. We believe that Orthogonal Transformer can serve as a strong general baseline and considerably advance a broad range of vision tasks. The idea of orthogonal self-attention may inspire researchers to explore efficient and effective attention mechanisms for high-dimensional data, such as high-definition images and long videos.

---

### Comment · Area_Chair_F7eu · 2022-08-09
**Further feedbacks from reviewers**

Dear authors,

Below please see the discussions from reviewers after the rebuttal. In general, the majority of the reviewers still have concerns that your orthogonal attention's contribution might be mixed with other architecture tweaks. Additional experiments are expected from them. Could you please state your plan or show some further quick results (if feasible) on how to better isolate the contribution of your orthogonal attention layer.

Reviewer hQkC:
I think the paper has an impressive system-level performance by mixing up with many other architecture tweaks. However, the ablation and analysis of the paper's own contribution have not been well demonstrated. The response from the author has only addressed part of my concerns.

I intend to raise my score to 5 as I believe the proposed Orth SA is somewhat new to the community and may benefit the research of SA mechanism. But I strongly encourage the authors to add more content to analyze the proposed method to strengthen their contributions.

Reviewer SBMR:
Overall, I think the orthogonal attention layer is a neat trick, and the experiments are pretty extensive. I still have some concern that this change is mixed up with many other architectural tweaks that mask the value of this component.

I think on balance this paper meets the bar for me. The Orth SA layer could be something more widely used. I see many concerns about its novelty/value of the Orth SA layer; if these are not addressed by the rebuttal and ablations (e.g. Table I, Appendix D), then I would not fight strongly for acceptance.

Reviewer 2jJ5:
I think the paper overall presents a system-level ViT variants to achieve its current performance. Regarding each of its core components, novelty is limited. The core component it claimed (orthogonal. sa) only gives about 0.3% top-1 imagenet improvement, as shown in Table 6, which is a big concern for me. I'm leaning towards borderline accept if authors can add more studies regarding the orthogonal operation or rephrase its contributions. Would not fight for acceptance as well if other reviewers still have unsolved concerns.

---

> ### Author Response · Authors · 2022-08-09
> **Plan to better isolate the contribution of the orthogonal attention layer**
>
> We sincerely thank the Area Chair and the reviewers for your efforts and valuable comments. We thank you for recognizing the positive aspects of our paper, such as the novelty of the orthogonal attention, the extensiveness of experiments, and the impressive performance of our models.
>
> The major concern is that the orthogonal attention's contribution might be mixed with other architecture tweaks. We would like to highlight that as shown in Table 6 and Table I, based on the same backbone except for the attention layers, the orthogonal attention can achieve better performance than other efficient self-attention mechanisms.
> To better isolate its contribution, we would like to compare the self-attention mechanisms on two additional backbones, including vanilla ViT and vanilla Swin Transformer. We will also compare the performance gain by the orthogonal attention among different backbones (Ortho/ViT/Swin). This might further validate whether or how much the orthogonal attention's contribution is mixed with other architecture tweaks. For time constraints, additional experiments will be included in the revised version.

---

### Meta-Review · Area_Chair_F7eu · 2022-08-25

**Recommendation:** Accept
**Confidence:** Less certain

**Metareview:**

The paper presents orthogonal attention mechanism for vision transformers. All reviewers found the overall system has good performance and the introduced orthogonal attention has the potential to be widely used. The authors' rebuttal resolves the majority of the questions.

The authors should add their promised additional experiments in the final version.

**Award:**

No

---

### Decision · Program_Chairs · 2022-09-14

Accept